# GSK3α phosphorylates dynamin-2 to promote GLUT4 endocytosis in muscle cells

Jessica Laiman[1], Yen-Jung Hsu[1], Julie Loh[1], Wei-Chun Tang[2], Mei-Chun Chuang[1], Hui-Kang Liu[3,4], Wei-Shun Yang[5], Bi-Chang Chen[2], Lee-Ming Chuang[1,6], Yi-Cheng Chang[6,7,8], and Ya-Wen Liu[1,9]

**Insulin-stimulated translocation of glucose transporter 4 (GLUT4) to plasma membrane of skeletal muscle is critical for postprandial glucose uptake; however, whether the internalization of GLUT4 is also regulated by insulin signaling remains unclear. Here, we discover that the activity of dynamin-2 (Dyn2) in catalyzing GLUT4 endocytosis is negatively regulated by insulin signaling in muscle cells. Mechanistically, the fission activity of Dyn2 is inhibited by binding with the SH3 domain of Bin1. In the absence of insulin, GSK3α phosphorylates Dyn2 to relieve the inhibition of Bin1 and promotes endocytosis. Conversely, insulin signaling inactivates GSK3α and leads to attenuated GLUT4 internalization. Furthermore, the isoform-specific pharmacological inhibition of GSK3α significantly improves insulin sensitivity and glucose tolerance in diet-induced insulin-resistant mice. Together, we identify a new role of GSK3α in insulin-stimulated glucose disposal by regulating Dyn2-mediated GLUT4 endocytosis in muscle cells. These results highlight the isoform-specific function of GSK3α on membrane trafficking and its potential as a therapeutic target for metabolic disorders.**

## Introduction

Skeletal muscle is the primary responsive tissue for insulin-stimulated glucose uptake (Kraegen et al., 1985) and is also the earliest tissue to exhibit insulin resistance, preceding the development of type 2 diabetes (T2D; Rothman et al., 1995; Warram et al., 1990). The insulin-induced glucose uptake is executed by the glucose transporter GLUT4, specifically expressed in muscle and adipose tissues (Birnbaum, 1989; Charron et al., 1989; James et al., 1989). Upon insulin stimulation, the insulin receptor–phosphoinositide 3-kinase (PI3K)-Akt signaling cascade is activated to mediate translocation of GLUT4 from intracellular membrane compartments to the cell surface, including sarcolemma and T-tubules in skeletal muscle, thus reducing blood glucose (Kohn et al., 1996; Tremblay et al., 2001; Wang et al., 1999). Contrary to its role in promoting GLUT4 exocytosis, the role of the insulin signaling pathway in GLUT4 endocytosis remains elusive.

A recent phosphoproteomic study demonstrated that dysregulation of vesicle trafficking is one of the fundamental defects of insulin resistance in the muscle cells of T2D patients (Batista et al., 2020), underscoring the importance of further exploration of GLUT4 vesicular trafficking in the pathogenesis of insulin

resistance. The GLUT4 is internalized mainly through dynamin-2 (Dyn2)–mediated endocytosis. Dyn2 is a large GTPase with a well-known function in catalyzing membrane fission, the last step of various routes of endocytosis (Ferguson and De Camilli, 2012; McMahon and Boucrot, 2011; Mettlen et al., 2009). To drive endocytosis, Dyn2 needs to be recruited to plasma membrane by interacting with the SH3 domain of many BAR domain containing proteins, such as Bin1, through its proline–arginine rich domain (PRD; Fig. 1 A), where these proteins generate a highly curved membrane to facilitate Dyn2 assembly and fission activity (Chappie et al., 2010; Faelber et al., 2011; Grabs et al., 1997; Zheng et al., 1996).

Dyn2 is ubiquitously expressed in mammals, yet mutations of Dyn2 lead to a tissue-specific disease, autosomal dominant centronuclear myopathy (CNM; Bitoun et al., 2005; Zhao et al., 2018). Previous studies have demonstrated that CNM-associated mutants of Dyn2 are hyperactive due to abnormal self-assembly (Chin et al., 2015; Kenniston and Lemmon, 2010; Wang et al., 2010). Expression of these Dyn2 mutants causes T-tubules, specialized structures derived from sarcolemma invaginations, to be abnormally shaped in mice (Cowling et al., 2011) and

...........................................................................................................................................................................................................................................................................................................
[1]Institute of Molecular Medicine, College of Medicine, National Taiwan University, Taipei, Taiwan; [2]ResearchCenter for Applied Sciences, Academia Sinica, Taipei, Taiwan; [3]National Research Institute of Chinese Medicine, Ministry of Health and Welfare, Taipei, Taiwan; [4]Program in the Clinical Drug Development of Herbal Medicine, Taipei Medical University, Taipei, Taiwan; [5]Division of Nephrology, Department of Internal Medicine, National Taiwan University Hospital, Hsin-Chu Branch, Hsin-Chu, Taiwan; [6]Department of Internal Medicine, National Taiwan University Hospital, Taipei, Taiwan; [7]Institute of Medical Genomics and Proteomics, College of Medicine, National Taiwan University, Taipei, Taiwan; [8]Institute of Biomedical Sciences, Academia Sinica, Taipei, Taiwan; [9]Center of Precision Medicine, College of Medicine, National Taiwan University, Taipei, Taiwan.

Correspondence to Ya-Wen Liu: yawenliu@ntu.edu.tw; Yi-Cheng Chang: yichengchang@ntu.edu.tw.

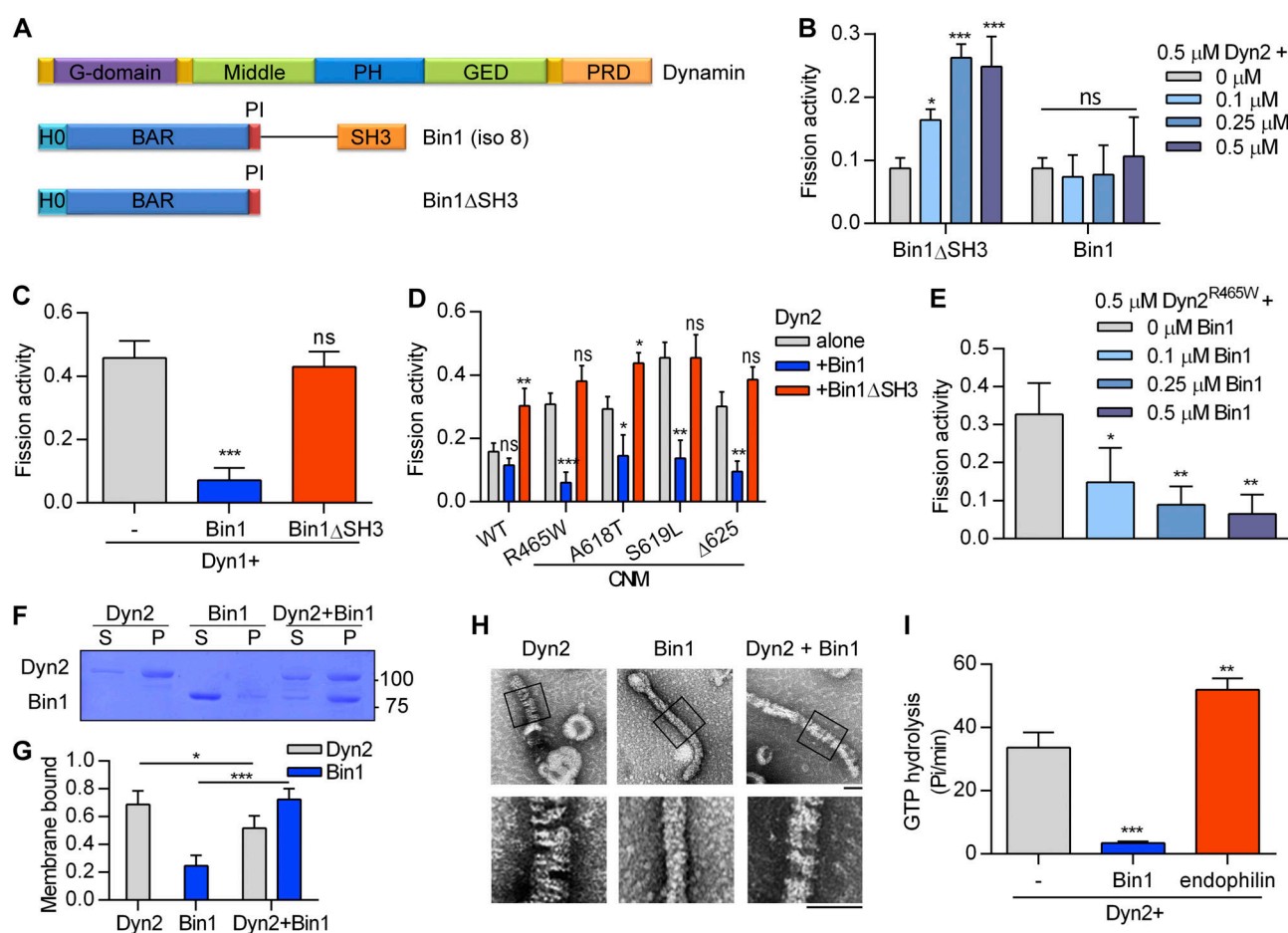

Figure 1. **Bin1 inhibits Dyn2 fission activity via SH3 domain. (A)** Domain structure of dynamin and Bin1 constructs used in this study. Pleckstrin homology domain and GTPase effector domain of dynamin are abbreviated to PH and GED, respectively. **(B)** In vitro fission assay of Dyn2. Purified Dyn2 was incubated with increasing concentrations of Bin1 or Bin1ΔSH3 in the presence of GTP and SUPER template. The fission activity was measured as the fraction of vesicle released from total SUPER template ($n = 3$ experimental replicates). **(C–E)** Effect of Bin1 on the fission activity of dynamins. Purified Dyn1 (C) or Dyn2 CNM mutants (D) is incubated with Bin1, Bin1ΔSH3, or increasing concentrations of Bin1 (E; $n = 3$ experimental replicates). **(F and G)** Liposome binding ability. Dyn2 or Bin1 were incubated with 100 nm liposome for 10 min at 37°C. Liposome-bound proteins (pellet, P) were separated from unbound ones (supernatant, S) with centrifugation sedimentation. The fraction of liposome-bound proteins was quantified and shown in G ($n = 3$ experimental replicates). Molecular weight is in kD. **(H)** Assembly of Dyn2 or Bin1 on liposome. Dyn2 and/or Bin1 were incubated with liposomes and then visualized with negative-stain TEM. Scale bar, 100 nm. Boxed areas were magnified and shown as below. **(I)** Liposome-stimulated GTPase activity of Dyn2. Dyn2 was incubated alone or with Bin1 or endophilin in the presence of liposome and GTP at 37°C. The rate of GTP hydrolysis was measured using a colorimetric malachite green assay ($n = 3$ experimental replicates). Data are shown as average ± SD and analyzed with one-way ANOVA. *$P < 0.05$; **$P < 0.01$; ***$P < 0.001$. Source data are available for this figure: SourceData F1.

zebrafish (Gibbs et al., 2014), or fragmented in *Drosophila* (Chin et al., 2015), suggesting excessive membrane fission of T-tubules due to Dyn2 hyperactivity.

Besides Dyn2, mutations of its binding partner Bin1/amphiphysin 2 also result in autosomal recessive CNM, suggesting they are loss-of-function mutations (Böhm et al., 2014; Nicot et al., 2007). Bin1 contains an N-BAR domain that forms a banana-shaped homodimer to sense, coordinate, and generate membrane curvature, and is required for biogenesis and maintenance of T-tubules in skeletal muscles (Al-Qusairi and Laporte, 2011; Lee et al., 2002; Razzaq et al., 2001). Prior research revealed that the downregulation of Dyn2 could rescue the phenotypes of Bin1-related CNM mouse model (Cowling et al., 2017). In contrast, increasing Bin1 expression ameliorates Dyn2-associated myopathy in mice (Lionello et al., 2022). These

findings suggest a role for Bin1 as a negative regulator of Dyn2. However, the detailed molecular mechanism of how Bin1 and Dyn2 interact to regulate endocytosis in muscle remains largely unknown.

Glycogen synthase kinase 3 (GSK3) is a serine/threonine kinase initially identified as a key regulator of glycogen synthase, suppressing glycogen synthesis via phosphorylation (Embi et al., 1980; Woodgett and Cohen, 1984). GSK3 is negatively regulated by insulin-PI3K-Akt signaling through phosphorylation of its N-terminal serine residue (Cross et al., 1995; Stambolic and Woodgett, 1994). Importantly, higher GSK3 expression and activity have been observed in the skeletal muscle of T2D patients and diabetic mice (Eldar-Finkelman and Krebs, 1997; Nikoulina et al., 2000). There are two ubiquitously expressed forms of GSK3 in mammals, GSK3α and GSK3β, which share high

similarity, but are not functionally redundant (Woodgett, 1990). GSK3β deletion in mice results in embryonic lethality, whereas GSK3α deficient mice are viable and display enhanced insulin sensitivity (Hoeflich et al., 2000; MacAulay et al., 2007). Most previous research focused on GSK3β, but emerging evidence demonstrated the isoform-specific effects of GSK3α on metabolic, cardiovascular, and neurodegenerative disorders (Ahmad et al., 2014; Ma, 2014; Nakamura et al., 2019). In particular, GSK3α has been implied to have a specific function in endocytosis to mediate synaptic plasticity in the hippocampus (Draffin et al., 2021).

Here, we demonstrate that Dyn2 inhibition by Bin1 is relieved by phosphorylation of Dyn2-PRD by GSK3α, which reduces the binding between the SH3 domain of Bin1 and Dyn2-PRD. Insulin signaling-mediated inhibition of GSK3α attenuates the fission activity of Dyn2 and reduces GLUT4 endocytosis in muscle cells. Furthermore, a GSK3α isoform-specific inhibitor improves insulin sensitivity and signaling in diet-induced insulin-resistant mice.

## Results

### Bin1 negatively regulates the fission activity of Dyn2

To measure the membrane fission activity of Dyn2, we quantified Dyn2-catalyzed vesicles released from the supported bilayer with excess membrane reservoir (SUPER) templates in vitro (Pucadyil and Schmid, 2008). We investigated the effect of Bin1 on Dyn2 and found that the N-BAR domain of Bin1 (Bin1ΔSH3) enhanced Dyn2-mediated membrane fission in a concentration-dependent manner. By contrast, full-length Bin1 lost this effect (Fig. 1 B). To test whether Bin1 inhibits the fission activity of dynamin, we examined its effects on membrane fission activity of Dyn1 and CNM mutants of Dyn2, including R465W, A618T, S619L, and Δ625, which possess higher membrane fission activity than WT-Dyn2 (Chin et al., 2015; Liu et al., 2011). As expected, these proteins exhibited greater fission activity than WT-Dyn2 (Fig. 1, C and D) and were inhibited by Bin1 in a dose-dependent manner (Fig. 1 E).

The inhibitory effect of Bin1 on Dyn2 does not arise from interfering with membrane binding of Dyn2, as revealed by liposome binding assays in which a substantial amount of Dyn2 remained bound to the membrane in the presence of Bin1 (Fig. 1, F and G). This result suggests the coassembly of Dyn2 with Bin1 on the membrane may alter Dyn2 activity. To further visualize how Dyn2 and Bin1 assemble on membrane, we performed negative-stain transmission electron microscopy (TEM) to observe liposomes incubated with purified Dyn2 or Bin1. Dyn2 assembly on liposomes resulted in membranous tubule formation decorated with helical-assembled Dyn2, a prerequisite for Dyn2 fission activity (Fig. 1 H). The average diameter of Dyn2-decorated tubes is 44.5 ± 2.3 nm with a spiral appearance regularly spaced with pitch of 8.46 ± 2.31 nm (Fig. S1, A and B). Assembly of Bin1 on liposomes also induced the formation of membrane tubules of a smaller diameter (31.4 ± 1.2 nm) that lack the regular spacing pattern. Meanwhile, coincubation of Dyn2 and Bin1 led to fewer membrane tubules of an intermediate diameter (37.4 ± 3.7 nm) with less-ordered, spiral-like protein assembly with higher pitch (18.35 ± 8.10 nm), implying an

altered assembly pattern. This is consistent with the result that Bin1 inhibits Dyn2 GTPase activity stimulated by liposomes (Fig. 1 I). By contrast, endophilin, another SH3 domain–containing protein that binds Dyn2, promoted the GTPase activity of Dyn2 in liposome-stimulated GTPase assays (Fig. 1 I), indicating that distinct SH3 domains have differential regulatory effects on Dyn2. These results reveal the dual roles of Bin1 on Dyn2 fission activity governed by its N-BAR and SH3 domains; in the absence of SH3 domain, N-BAR stimulates Dyn2, and the inclusion of SH3 inhibits membrane fission activity of Dyn2 by altering Dyn2 assembly on the membrane.

### The PI motif of Bin1 is crucial for membrane binding and tubulation, but not Dyn2 fission activity

The muscle-specific isoform of Bin1 contains a unique positively charged, 15-amino-acid stretch called the PI motif that enhances the lipid-binding ability of Bin1 (Lee et al., 2002). We tested the potential role of PI motif in Dyn2-mediated membrane fission using different Bin1 variants (Fig. S1 C). To our surprise, PI motif did not affect Bin1 regulation on Dyn2 fission activity (Fig. S1 D). Nonetheless, consistent with a previous report (Lee et al., 2002), the PI motif was essential for the induction of plasma membrane tubule formation (tubulation) in cells (Fig. S1 E). These results suggest that the PI motif is important for membrane binding and tubulation of Bin1, i.e., the T-tubule biogenesis, yet is dispensable for regulating Dyn2 fission activity.

### CNM-associated Bin1 with SH3 domain mutation enhances Dyn2 fission activity

Among the CNM-related Bin1 mutations, Bin1$^{Q434X}$ and Bin1$^{K436X}$ contain premature stop codons that truncate the SH3 domain (Fig. 2 A; Böhm et al., 2010; Nicot et al., 2007). We speculated that these CNM-Bin1 mutants may lose the inhibitory effect on Dyn2. Indeed, GST pulldown assay showed significant decrease in the binding of Bin1$^{Q434X}$ and Bin1$^{K436X}$ to GST-Dyn2-PRD (Fig. 2, B and C), while displaying membrane binding and tubulation activities similar to those of Bin1$^{WT}$ (Fig. 2, D and E). Compared with Bin1$^{WT}$, both Bin1 mutants with SH3 domain truncation significantly increased Dyn2 fission activity in vitro (Fig. 2 F), and their coincubation with Dyn2 generated liposome tubules with smaller diameter (Fig. 2, G and H). These data support the idea that Bin1-SH3 inhibits Dyn2 and raises the possibility that enhanced Dyn2 activity could be part of the pathogenesis of CNM associated with Bin1 mutations.

Consistent with the result from liposome tubulation (Fig. 2 E), the expression of GFP-tagged Bin1$^{Q434X}$ and Bin1$^{K436X}$ in the C2C12 myoblasts induced tubulation of the plasma membrane similar to that by Bin1$^{WT}$-GFP (Fig. 2 I). The morphology of these membrane tubules generated by Bin1-GFP was reminiscent of T-tubules in terms of their high curvatures and the enrichment of Bin1 and PI(4,5)P2 (Lee et al., 2002). They could be severed into small vesicles by Dyn2, making them a feasible cellular system to validate Bin1-Dyn2 interactions observed in the reconstitution experiments (Chin et al., 2015; Laiman and Liu, 2020). We thus transfected both Dyn2-mCherry and Bin1-GFP into C2C12 myoblasts and examined the morphology of membrane tubules (Fig. 2 J). We categorized the cells according to the

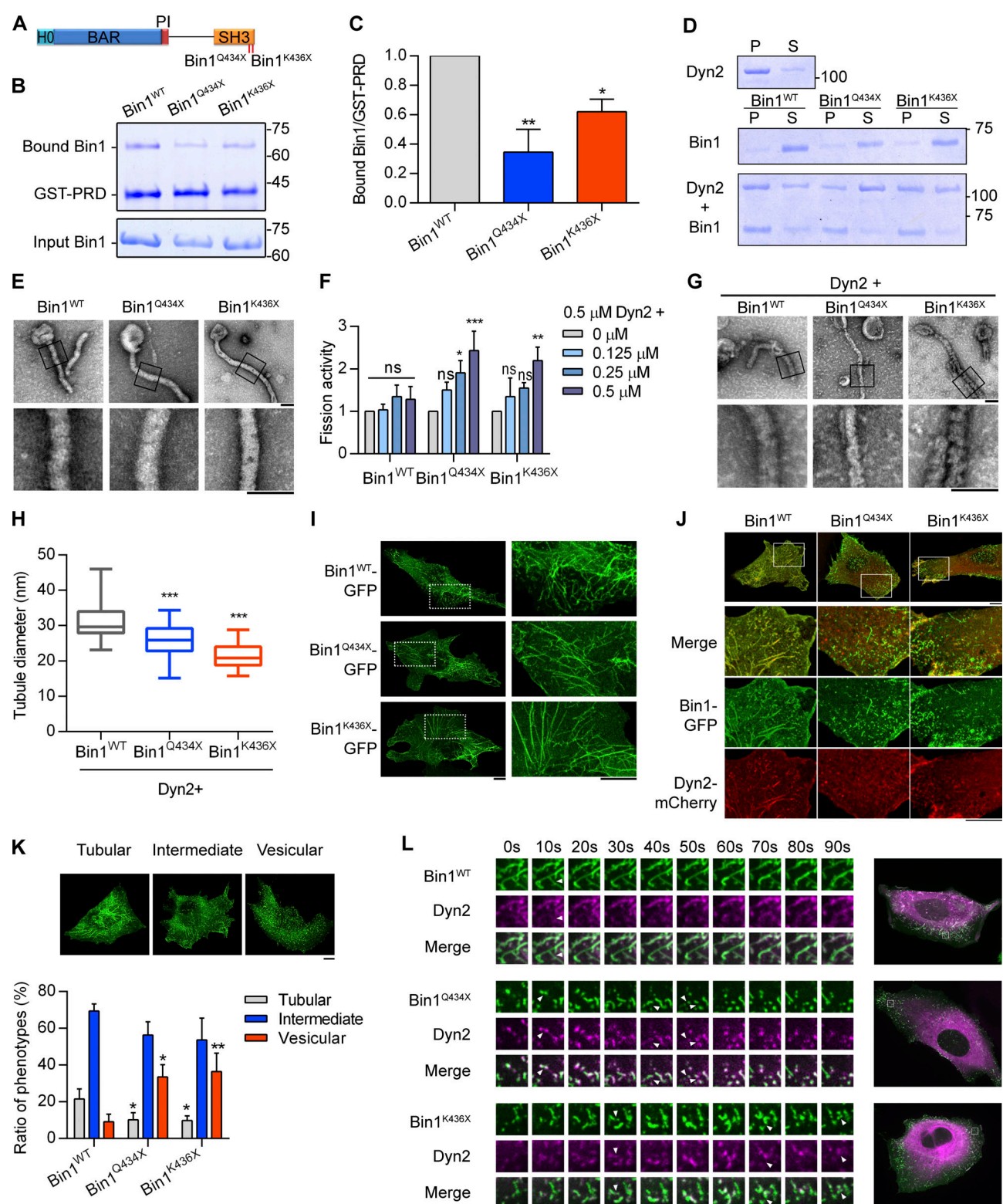

Figure 2. **CNM-Bin1 mutants enhance Dyn2 fission activity. (A)** CNM-associated Bin1 mutants used in this study. **(B and C)** Binding ability between Bin1 mutants and the PRD of Dyn2. GST-PRD was incubated with indicated purified His-tagged Bin1 in the presence of liposome. PRD-bound Bin1 was analyzed with SDS-PAGE and Coomassie Blue staining. The ratio of bound Bin1 was normalized to WT and shown in C ($n$ = 3 experimental replicates). **(D)** Membrane-binding ability of CNM-Bin1-SH3 mutants. 0.5 µM Bin1 mutants and/or 0.5 µM Dyn2 were incubated with 100 nm liposome for 10 min at 37°C. Liposome-bound proteins (pellet, P) were separated from unbound ones (supernatant, S) through centrifugation sedimentation. **(E)** Assembly of Bin1 mutants on liposome. 1 µM Bin1 was incubated with liposome and then visualized with negative-stain TEM. Scale bar, 100 nm. Boxed areas were magnified and shown as below. **(F)** Effects of CNM-Bin1 on Dyn2 fission activity. Dyn2 was incubated with increasing concentrations of Bin1[WT], Bin1[Q434X], or Bin1[K436X] in the presence of GTP and SUPER template. The fission activity was determined by sedimentation and fraction of released fluorescent vesicles in the supernatant from total SUPER

template ($n$ = 3 experimental replicates). **(G and H)** Electron micrographs of Dyn2 together with Bin1 mutants assembled onto liposomes. WT Dyn2 and Bin1 mutants were incubated with 100 nm liposomes for 10 min at 37°C, adsorbed to grids, and imaged by negative-stain TEM. Scale bar, 100 nm. Boxed areas were magnified and shown. Averaged tubule diameter was quantified and shown in H ($n \geq 7$ liposome tubules). **(I)** Membrane tubulation ability of Bin1 mutants in cellulo. GFP-tagged Bin1 mutants were overexpressed in C2C12 myoblasts through transfection and imaged under confocal microscopy. Scale bar, 10 μm. **(J and K)** Morphology of CNM-associated Bin1 mutants coexpressed with Dyn2-mCherry in myoblast. GFP-tagged WT or mutant Bin1 was transfected into C2C12 myoblasts together with Dyn2-mCherry. The Bin1-mediated membrane morphologies were imaged with confocal microscopy. Bottom panels, magnified images from insets in top panels. Scale bar, 10 μm. The Bin1-mediated membrane deformation was categorized into three groups (the examples of cells from each category are shown in top panel of K), and the ratio of each population was quantified and compared with Bin1$^{WT}$ ($n \geq 75$ cells from three independent repeats). **(L)** Time-lapse representative images of CNM-associated Bin1 mutants coexpressed with Dyn2-mCherry in C2C12 myoblast were magnified and shown. White arrowheads indicate the occurrence of membrane fission. Scale bar, 2 μm. Data are shown as average ± SD and analyzed with one-way ANOVA. *P < 0.05; **P < 0.01; ***P < 0.001. Molecular weight is in kD. Source data are available for this figure: SourceData F2.

overall Bin1-GFP tubulation patterns (Fig. 2 K). Bin1$^{Q434X}$ and Bin1$^{K436X}$ coexpression with Dyn2 significantly reduced the proportion of cells with Bin1 tubules, with a reciprocal increase in cells that showed the characteristic vesicular membrane patterns (Fig. 2 K). Furthermore, live-cell imaging of the dynamics of Bin1-mediated tubules revealed more frequent fission events in the presence of Bin1$^{Q434X}$ or Bin1$^{K436X}$ (Video 1 and Fig. 2 L, arrowheads). Together, these data suggest that CNM-associated Bin1 mutants with truncated SH3 domain have weakened inhibition on Dyn2, underscoring the importance of Bin1-SH3 in tuning Dyn2 fission activity.

### Phosphorylation of Dyn2-PRD confers resistance to Bin1 inhibition and promotes endocytosis

The above data on CNM-associated Bin1 mutations indicated that Dyn2-PRD could be a regulatory hub for Dyn2 activity under physiological contexts. Given that phosphorylation of Dyn1-PRD has been known to regulate Dyn1 function (Clayton et al., 2010; Reis et al., 2015; Slepnev et al., 1998), we hypothesized that similar mechanisms operate in Dyn2-PRD. Among the putative phosphorylation sites on Dyn2-PRD, we focused on Ser848 and Ser856, two serine residues located within the SH3 binding pockets of Dyn2-PRD (Fig. 3 A; Choudhary et al., 2009; Efendiev et al., 2002). Using GST pulldown assays, we found that Dyn2$^{S848E}$, a phosphomimetic mutant of Dyn2-S848, had decreased binding affinity to Bin1-SH3, but not to endophilin-SH3 (Fig. 3, B and C, and Fig. S2, A–C). In vitro fission assays and analysis of Bin1-GFP tubule in Dyn2-cotransfected C2C12 myoblasts further confirmed that the membrane fission activity of Dyn2$^{S848E}$ was increased by Bin1 (Fig. 3, D–F, and Fig. S2 D; and Video 2). Moreover, Bin1 and Dyn2$^{S848E}$ assembled into smaller tubules on liposomes similar to those of Dyn2$^{WT}$ and Bin1$^{Q434X}$ or Bin1$^{K436X}$ (Fig. 3 G and Fig. S2 E). These results suggested that Dyn2-S848 is a critical residue governing Dyn2-Bin1 interaction, likely through its phosphorylation status. Of note, the S848 residue is evolutionarily conserved among DNM2 genes from different vertebrates, whereas Dyn2-S856 residue is not found in zebrafish (Fig. S2 F).

To measure the effect of Dyn2$^{S848}$ phosphorylation on endocytosis in muscle cells, we investigated the role of Dyn2$^{S848}$ phosphorylation on GLUT4 endocytosis by performing subcellular fractionation of C2C12 myotubes that express high amount of Bin1 (Fig. S3 A). C2C12 myotubes with Dyn2$^{S848A}$ overexpression had a higher ratio of GLUT4 in plasma membrane than those infected with Dyn2$^{S848E}$ and Dyn2$^{WT}$, consistent with

reduced Dyn2 activity due to Bin1-SH3 inhibition (Fig. 4, A and B). Of note, the total amount of GLUT4 remained similar in cells expressing WT or mutants Dyn2 (Fig. S3, B and C). GLUT4 membrane translocation is stimulated by insulin through the PI3K-Akt signaling cascade in muscle cells (Beg et al., 2017; Katome et al., 2003; Leto and Saltiel, 2012). To further validate the physiological importance of Dyn2-S848 phosphorylation, we utilized L6 myoblasts to investigate the effect of Dyn2$^{S848}$ phosphorylation on the internalization of GLUT4-GFP with an extracellular HA epitope (HA-GLUT4-GFP) triggered by serum starvation (SS; Lampson et al., 2000). Quantification of cell surface GLUT4 by immunostaining without permeabilization showed that GLUT4 displayed a predominant intracellular distribution after 3 h of SS in Dyn2$^{WT}$-expressing cells (Fig. 4, C–E). Intriguingly, Dyn2$^{S848A}$ overexpression resulted in significantly higher HA-GLUT4-GFP level on the cell surface upon SS, whereas Dyn2$^{S848E}$ displayed a lower surface level of GLUT4 even with the growth medium containing 10% FBS, indicating that Dyn2$^{S848}$ phosphorylation facilitates endocytosis in the muscle cells, whereas phospho-resistant Dyn2$^{S848A}$ blocks GLUT4 internalization.

We also monitored the internalization of Alexa488-labeled transferrin (Tfn-488). A Dyn2 mutant with a curvature-generation defect, Dyn2$^{G537C}$, served as a negative control and displayed lower transferrin internalization than Dyn2$^{WT}$. Similar to its effect on endogenous GLUT4 distribution, the overexpression of Dyn2$^{S848A}$ led to less transferrin internalization in C2C12 myotubes (Fig. S3, D and E). Of note, neither phosphoresistant nor phosphomimetic mutants of Dyn2-Ser848 affect the exocytic pathway, monitored by the efficiency of transferrin recycling (Fig. S3, F–H). These results demonstrate that phosphorylation of Dyn2-Ser848 enhances endocytosis in muscle cells.

### Phosphorylation of Dyn2-Ser848 is attenuated by insulin signaling

We next sought evidence of endogenous phosphorylation of Dyn2-S848 by immunoprecipitation and phosphoserine detection in L6 myoblasts. The pan-phosphoserine (p-Ser) immunoblotting experiment showed significantly elevated p-Ser in precipitated HA-Dyn2$^{WT}$ under SS but not in HA-Dyn2$^{S848A}$ (Fig. 5, A and B). To further pinpoint the phosphorylation site of Dyn2, we generated a specific antibody for phosphorylated Dyn2-Ser848 (named p-Dyn2$^{S848}$) and detected a higher p-Dyn2$^{S848}$ signal in C2C12 myotubes after SS (Fig. 5 C).

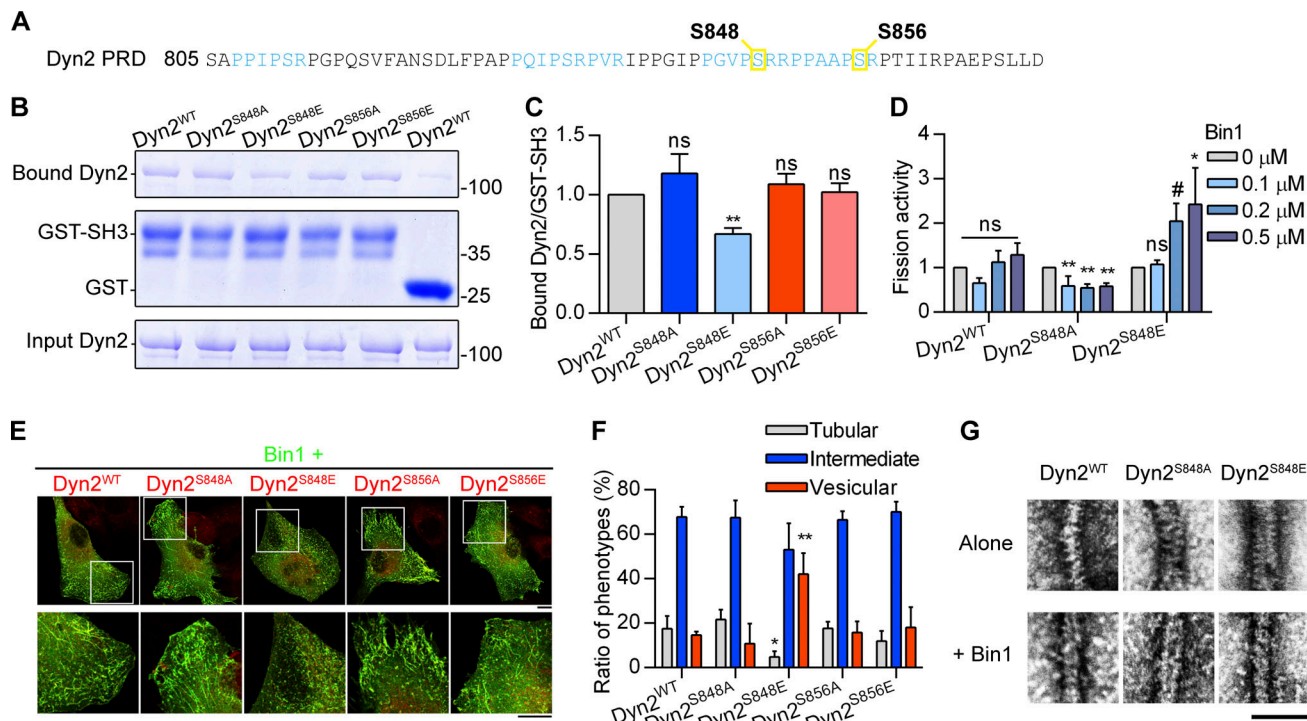

Figure 3. **The phosphorylation of Dyn2-S848 relieves Bin1 inhibition. (A)** The amino acid sequence of Dyn2-PRD. Two potential phosphorylated serine, S848 and S856, are boxed and highlighted in yellow. The SH3-binding pockets were highlighted in light blue. **(B and C)** GST pulldown assay. GST or GST-tagged Bin1 SH3 domain was incubated with indicated Dyn2 mutants. The ratio of bound Dyn2 was detected with Coomassie Blue staining, quantified with ImageJ, and normalized to WT as shown in C ($n$ = 3 experimental replicates). Molecular weight is in kD. **(D)** Fission activity of Dyn2 mutants in the presence of Bin1. Indicated Dyn2 proteins were incubated with increasing concentrations of Bin1 in the presence of GTP and SUPER templates for fission activity analysis. The data was shown as fold change relative to the Dyn2 fission activity in the absence of Bin1 ($n$ = 3 experimental replicates). **(E and F)** Effect of Dyn2 mutants on GFP-Bin1 tubules in myoblasts. WT or indicated Dyn2-mCherry mutants were transfected into myoblasts together with Bin1-GFP. Scale bar, 10 μm. The Bin1-induced membrane morphologies were analyzed as in Fig. 2 K and shown in F ($n$ ≥ 100 cells from three independent repeats). **(G)** Assembly of Dyn2 mutants on liposome. Indicated Dyn2 proteins were incubated with liposome alone (top) or together with Bin1 (bottom) and then visualized with negative-stain TEM. Scale bar, 100 nm. Data are shown as average ± SD and analyzed with one-way ANOVA. #P ≤ 0.06; *P < 0.05; **P < 0.01. Source data are available for this figure: SourceData F3.

Importantly, the phosphorylation of Dyn2-S848 was significantly increased by SS and reduced by insulin (Fig. 5, D and E). Similar to the effect of Dyn2S848E on Bin1-GFP tubules, SS induced the fragmentation of Bin1-GFP tubules in myoblasts cotransfected with Dyn2, and this effect could be reversed by insulin (Fig. 5, F and G).

## GSK3α, but not GSK3β, phosphorylates Dyn2 at Ser848 and promotes endocytosis

Using Scansite 4.0 to predict possible kinases of Dyn2S848, we identified GSK3α as a candidate for phosphorylating Dyn2-S848 (Fig. S4 A). GSK3α is a constitutively active serine/threonine kinase that is inhibited by insulin-PI3K-Akt signaling through phosphorylation at Ser21 (Cross et al., 1995; Frame et al., 2001). We thus tested whether GSK3α increases the phosphorylation of Dyn2-Ser848 by expressing HA-Dyn2 with either WT, constitutively active (S21A, GSK3α CA), or kinase-inactive (K148A, GSK3α KI) GSK3α in cells. GSK3α CA significantly enhanced p-Ser signal of HA-Dyn2WT without affecting the extent of phosphorylation of HA-Dyn2S848A (Fig. 6, A and B, and S4 B). GSK3β, which shares a high similarity in its catalytic domain with that of GSK3α, might also phosphorylate Dyn2-S848

(Woodgett, 1990). However, unlike GSK3α, GSK3β CA showed no effects on the phosphorylation status of S848 in HA-Dyn2WT and HA-Dyn2S848A (Fig. 6, C and D), consistent with a previous report that Dyn2 is not phosphorylated by GSK3α in vitro (Clayton et al., 2010). We further performed an in vitro kinase assay and demonstrated direct phosphorylation of Dyn2-Ser848 by recombinant GSK3α (Fig. 6, E and F), which was confirmed by mass spectrometry (MS) detection (Fig. S4 C).

Similar to the inhibitory effect of Dyn2S848A on transferrin endocytosis, the expression of the GSK3α KI resulted in reduced transferrin internalization in C2C12 myotubes (Fig. 7, A and B). GSK3α also affects GLUT4 distribution in L6 myoblasts, as shown by a lower ratio of cell surface HA-GLUT-GFP in cells expressing GSK3α CA (Fig. 7, C and D). We also tried to observe the effect of GSK3β, but the expression level of GSK3β CA and KI compared with that of their α counterparts was very low (Fig. S4 D); hence, we used isoform-specific inhibitors of GSK3α (BRD0705) or -β (TWS119) instead. BRD0705 is a recently developed small-molecule inhibitor of GSK3α (Wagner et al., 2018) that selectively inhibits GSK3α mainly by forming two hydrogen bonds with Glu196, a unique residue in its kinase hinge region that is not conserved in GSK3β (Fig. S5, A–C). GSK3α inhibition,

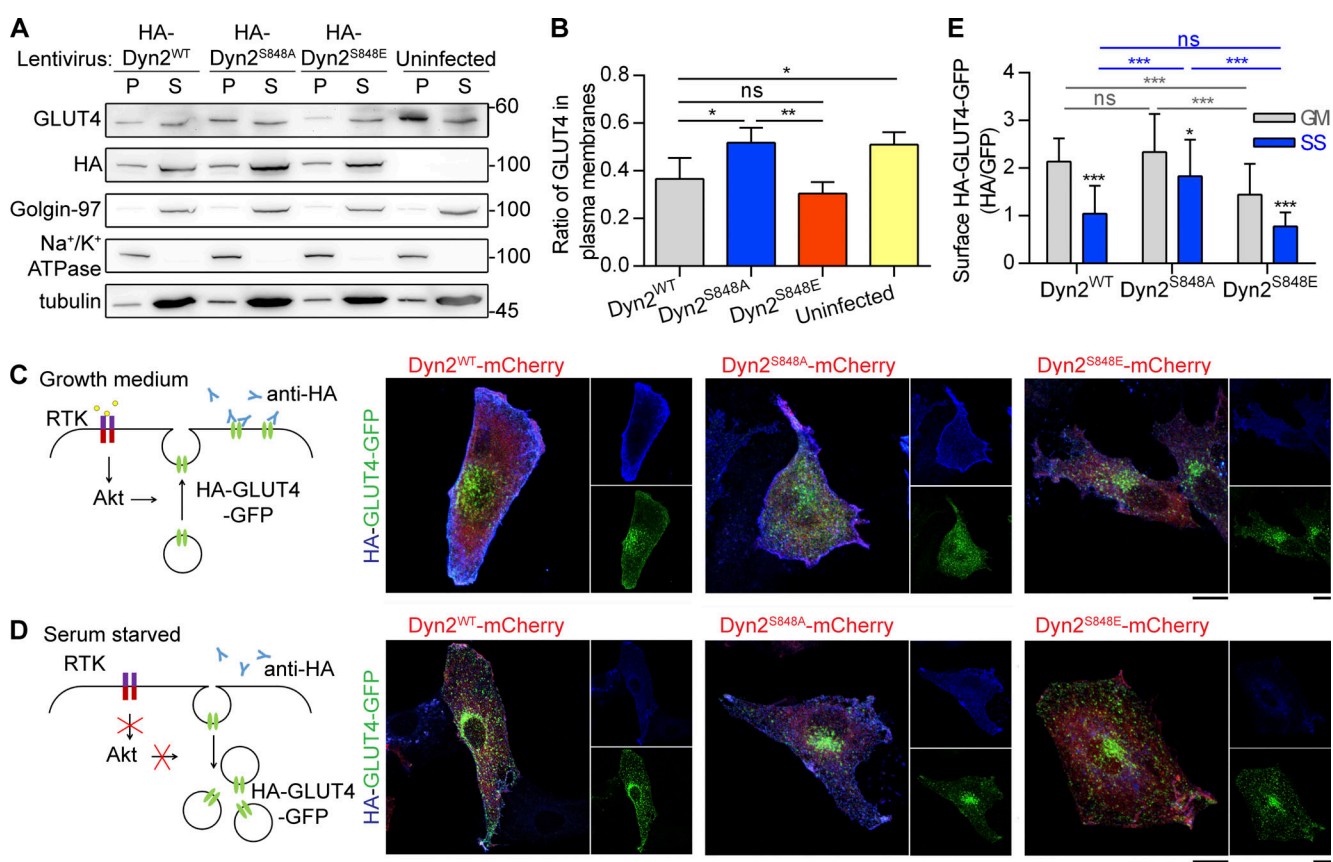

Figure 4. **The phosphorylation of Dyn2-S848 promotes endocytosis. (A and B)** Subcellular distribution of endogenous GLUT4 in C2C12 myotubes expressing different Dyn2 mutants. Differentiated C2C12 myotubes were infected with lentiviruses to express indicated HA-Dyn2 and then subjected to subcellular fractionation to determine the distribution of endogenous GLUT4 by Western blotting. The markers for heavy membrane (Na$^+$/K$^+$ ATPase, plasma membrane), light membrane (Golgin-97, trans-Golgi), and cytosol (tubulin) were used to validate this assay. The ratio of GLUT4 in heavy membrane fraction was quantified and shown in B ($n$ = 4 experimental replicates). Molecular weight is in kD. **(C–E)** Effects of Dyn2-S848 mutations on the insulin-regulated GLUT4-HA-GFP distribution. L6 myoblasts cotransfected with GLUT4-HA-GFP and Dyn2-mCherry mutants were subjected to growth or serum-free medium for 2 h. The cartoons illustrate the expected location of GLUT4-HA-GFP under indicated conditions; with or without receptor tyrosin kinase (RTK) signaling activation. After immunofluorescent staining with anti-HA antibody without permeabilization and imaging by confocal microscopy, the relative amount of surface GLUT4-HA-GFP was quantified by the fluorescent intensity of HA signaling in blue divided by the total GLUT4-HA-GFP in green as shown in E ($n$ ≥ 27 cells from three independent repeats). Data are shown as average ± SD and analyzed with one-way ANOVA. *P < 0.05; **P < 0.01; ***P < 0.001. Source data are available for this figure: SourceData F4.

but not that of GSK3β, resulted in a significantly higher ratio of HA-GLUT4-GFP on cell surface under both growth medium and SS (Fig. 7, E–H). Together, these data demonstrated that phosphorylation of Dyn2-Ser848 is a critical regulatory target for Bin1-SH3 and GSK3α that tunes the endocytic and membrane fission functions of Dyn2.

### Isoform-specific pharmacological inhibition of GSK3α improves insulin sensitivity in mice with high-fat high-sucrose diet (HFHSD)

Our findings of GSK3α isoform-specific role in regulating GLUT4 endocytosis in muscle cells led us to propose that the inhibition of GSK3α might improve insulin sensitivity in mice fed with HFHSD. 8-wk-old C57BL6/J mice were put on HFHSD and given BRD0705, the GSK3α inhibitor, (30 mg/kg/d) or vehicle by daily oral gavage since the age of 10 wk. There was no significant difference in body weight after administration of BRD0705 (Fig. 8 B). HFHSD-fed mice treated with BRD0705 showed

markedly improved glucose tolerance during glucose tolerance test (Fig. 8 C) and improved insulin sensitivity during insulin tolerance test (Fig. 8 D). We assessed the changes in phosphorylation of the Akt-GSK3-Dyn2 pathway in soleus muscle of mice treated with BRD0705 and vehicle with or without insulin injection after starvation (Fig. 8, E–I). Consistent with our observation in cultured myoblasts (Fig. 5 D), insulin significantly reduced Dyn2-S848 phosphorylation in vehicle-treated starved mice (Fig. 8 E). Compared with vehicle control, GSK3α inhibitor-treated starved mice exhibited significantly lower basal Dyn2-S848 phosphorylation. Furthermore, insulin reduced Dyn2-S848 phosphorylation in vehicle-treated mice but not in BRD0705-treated mice (Fig. 8 E). We also analyzed the phosphorylation status of insulin-signaling proteins, namely Akt, GSK3α, and GSK3β. GSK3α inhibition resulted in enhanced insulin-stimulated Akt$^{S473}$ phosphorylation (P = 0.065; Fig. 8 F). Consistently, GSK3α inhibition also increased insulin-stimulated phosphorylation of GSK3α$^{S21}$ significantly, and, to a much

Figure 5.  **The phosphorylation of Dyn2$^{S848}$ is regulated by insulin signaling. (A and B)** Phosphorylation of Dyn2 in response to SS. HA-tagged Dyn2$^{WT}$ or Dyn2$^{S848A}$ expressed in L6 myoblasts were precipitated by anti-HA antibody with or without SS. Phosphorylated Dyn2 was quantified by anti-phospho-serine antibody and normalized with precipitated HA intensity compared with HA-Dyn2$^{WT}$ in growth medium and shown in B ($n = 3$ experimental replicates). **(C–E)** Phosphorylation of endogenous Dyn2 in response to SS and insulin. C2C12 myotubes were subjected to growth medium (10% FBS), 3-h serum-free medium (SS; C), or SS followed with 30-min, 100 nM insulin stimulation (D). Cell lysates were harvested and detected by Western blotting with indicated antibodies. The ratio of phosphorylated Dyn2$^{S848}$ was quantified and shown in E ($n = 3$ experimental replicates). **(F)** Time-lapse representative images of Bin1-GFP tubules in cells cotransfected with Dyn2-mCherry and cultured in growth or serum-free medium. Boxed areas were magnified and shown in the lower panel. Scale bar, 10 μm. **(G)** Effects of insulin signaling on Bin1 tubule morphology. Bin1-GFP were transfected into myoblasts together with Dyn2-mCherry and cultured in growth medium, serum-starved, or starved followed by 60-min insulin stimulation. The population of cells with tubule, intermediate, or vesicular Bin1 morphology were quantified ($n \geq 75$ cells from three independent repeats). Scale bar, 10 μm. Data are shown as average ± SD and analyzed with one-way ANOVA. *$P < 0.05$; **$P < 0.01$; ***$P < 0.001$. Molecular weight is in kD. Source data are available for this figure: SourceData F5.

lesser degree, insulin-stimulated GSK3β$^{S9}$ phosphorylation (Fig. 8, G and H). These observations indicated that isoform-specific pharmacological inhibition of GSK3α improves insulin sensitivity in HFHSD-fed mice.

## Discussion

Activation of the insulin-Akt signaling cascade is known to trigger GLUT4 vesicle exocytosis to promote glucose uptake (Kohn et al., 1996; Tremblay et al., 2001; Wang et al., 1999). In this study, we reveal that insulin signaling also reduces GLUT4 endocytosis through GSK3α-mediated regulation of Dyn2 activity (Fig. 9). Mechanistically, insulin inactivates GSK3α, results in lower Dyn2

phosphorylation and the inhibition of Dyn2 by Bin1, and reduced GLUT4 internalization (Fig. 9, left). In the absence of insulin, active GSK3α phosphorylates Dyn2 to relieve the inhibition by Bin1 and facilitates GLUT4 endocytosis in muscle cells (Fig. 9, right). Given that both insulin and exercise could stimulate GLUT4 translocation (Richter and Hargreaves, 2013), GSK3α may exert an unappreciated role in contraction-stimulated glucose uptake in muscles. Furthermore, the integrative metabolic sensor AMP-activated protein kinase has been reported to reduce GLUT4 endocytosis in L6 myotubes and phosphorylate GSK3β at Ser9 (Fazakerley et al., 2010; Horike et al., 2008), suggesting a potential crossregulation between AMP-activated protein kinase and GSK3α for GLUT4 internalization in skeletal muscle.

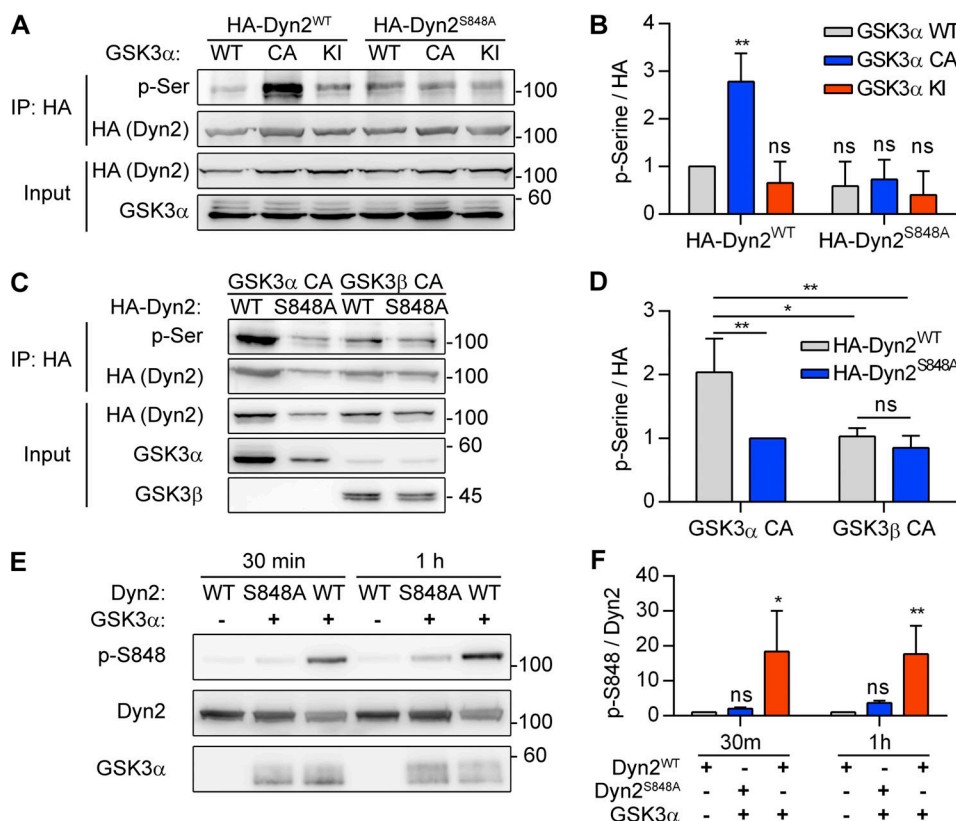

Figure 6. **GSK3α phosphorylates the S848 residue of Dyn2. (A and B)** Phosphorylation of Dyn2 in response to GSK3α overexpression. HA-Dyn2[WT] or HA-Dyn2[S848A] were coexpressed with WT, CA, or KI forms of GSK3α in Hela cells. After precipitation with anti-HA antibody, the phosphorylated Dyn2 was detected with anti-phospho-serine antibody, normalized with HA intensity, and shown in B ($n$ = 3 experimental replicates). **(C and D)** Isoform specific activity of GSK3 on Dyn2. HA-Dyn2[WT] or HA-Dyn2[S848A] were co-expressed with CA GSK3α or GSK3β in Hela cells. After precipitation with anti-HA antibody, the phosphorylated Dyn2 was detected with anti-phospho-serine antibody, normalized with HA intensity, and shown in D ($n$ = 3 experimental replicates). **(E and F)** In vitro kinase assay. 0.8 µg purified Dyn2[WT] or Dyn2[S848A] were incubated with ATP in the presence or absence of 10 ng purified GSK3α. After 30-min or 1-h incubation, the phosphorylated Dyn2 was quantified by Western blotting with anti–p-Dyn2[S848] antibody. The intensity of p-Dyn2[S848] was quantified with ImageJ, normalized with Dyn2 signal and shown in F ($n$ = 3 experimental replicates). Data are shown as average ± SD and analyzed with one-way ANOVA. *$P < 0.05$; **$P < 0.01$. Molecular weight is in kD. Source data are available for this figure: SourceData F6.

Dyn2 and Bin1 are interacting partners, and their mutations lead to the muscle-specific disease, CNM, despite their ubiquitous expression in human tissues (Bitoun et al., 2005; Böhm et al., 2014; Nicot et al., 2007). The tissue-specific pathogenicity of Bin1 and Dyn2 mutations underlines the importance of their regulations in muscle. Increased activity of Dyn2 in CNM-Bin1–expressing myoblasts with fragmented tubules is reminiscent of disrupted T-tubule network in mice expressing mutant Bin1 that lacks the SH3 domain (Fig. 2 J; Silva-Rojas et al., 2021). On the other hand, CNM-associated Dyn2 mutants display enhanced fission activity due to lacking autoinhibition, which leads to disrupted T-tubules in *Drosophila* and mouse muscles (Chin et al., 2015; Cowling et al., 2011). The phenotypic similarities between muscles that express CNM-Bin1 and CNM-Dyn2, and the tunable Dyn2-Bin1 interaction identified in this study suggested that Dyn2 hyperactivity is a common pathogenic mechanism of CNM, which may also lead to abnormal GLUT4 distribution in patients (González-Jamett et al., 2017). In line with that, both Dyn2 depletion and Bin1 overexpression could rescue CNM phenotype in several CNM mouse models, including myotubularin (MTM1; Cowling et al., 2014; Lionello

et al., 2019; Tasfaout et al., 2017). Further studies on how these three proteins physically and functionally interact in skeletal muscle would bring more insights into the pathological progression of CNM.

Mammalian GSK3 comprises two isoforms, GSK3α and GSK3β, that share high similarity but are not functionally redundant (Woodgett, 1990). Knockout of the sole GSK3 homolog in *Drosophila* could be rescued by overexpression of human GSK3β but not human GSK3α (Ruel et al., 1993). In mice, deletion of GSK3β results in embryonic lethality, while mice lacking GSK3α are viable and display enhanced insulin sensitivity, specifically in liver cells (Hoeflich et al., 2000; Kerkela et al., 2008; MacAulay et al., 2007). The PRD of Dyn1 has been shown to be phosphorylated by GSK3β (Clayton et al., 2010), and this modification inhibits Dyn1 activity on endocytosis (Clayton et al., 2009). Reduced Dyn1 inhibition by GSK3β in non-small lung cancer cells gives rise to rapid, dysregulated CME, also called adaptive CME (Reis et al., 2015). The isoform-specific regulation between GSK3β and Dyn1 is mirrored in our identification of Dyn2 as a specific substrate of GSK3α, but not GSK3β. More strikingly, GSK3α phosphorylation positively regulates

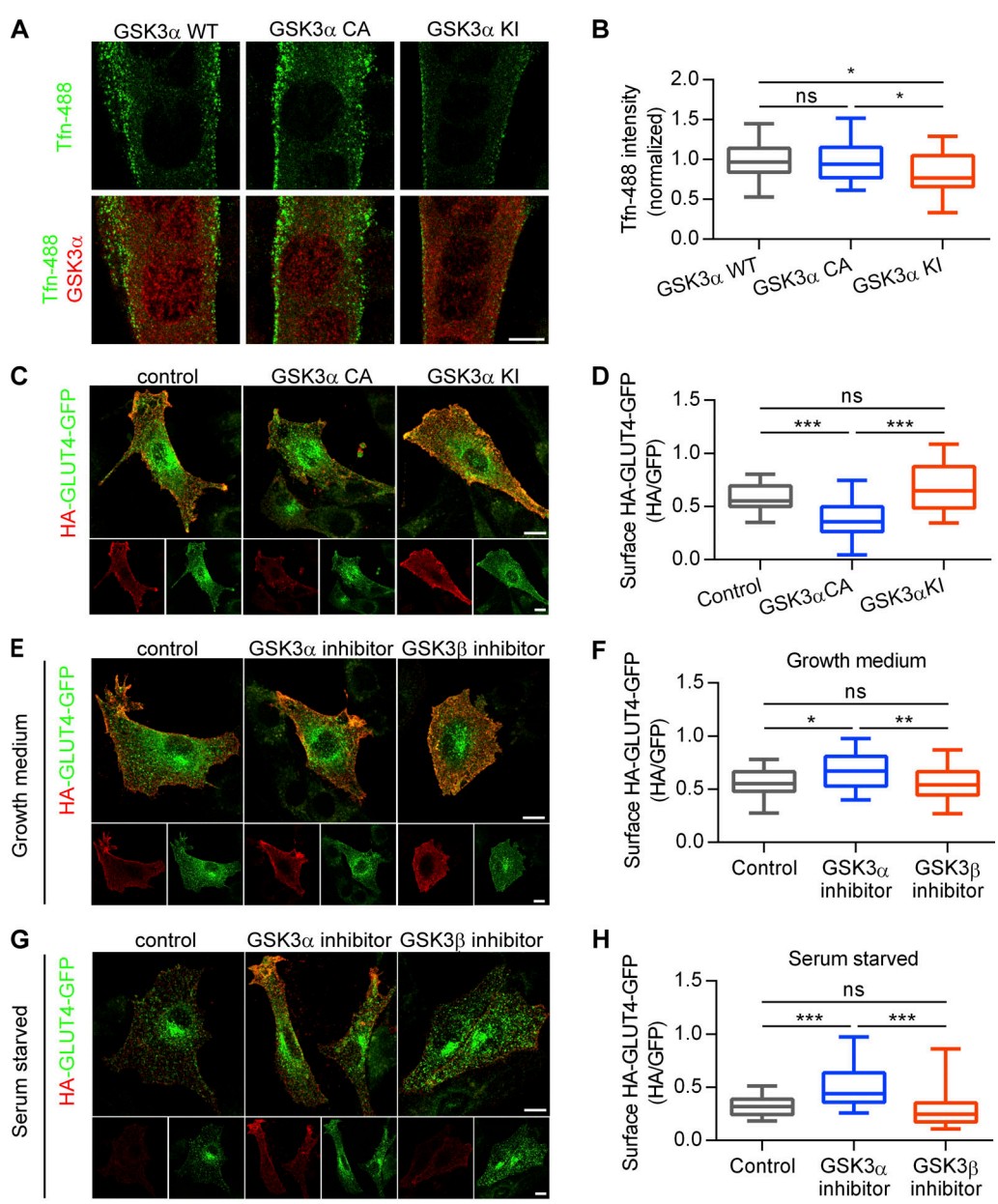

Figure 7. **GSK3α activation promotes endocytosis in muscle cells. (A)** Effect of GSK3α mutants on transferrin internalization in myotubes. C2C12 myoblasts transfected with indicated GSK3α mutants and differentiated into myotubes were incubated with Tfn-488 at 37°C for 20 min. Cells were then washed on ice, fixed, and imaged under confocal microscopy. Scale bar, 10 µm. **(B)** Fluorescence intensity of internalized transferrin was quantified and shown ($n = 25$ cells from three independent repeats). **(C)** Effect of GSK3α mutants on GLUT4-HA-GFP distribution. L6 myoblasts were infected with indicated GSK3α mutants and transfected with GLUT4-HA-GFP. Scale bar, 10 µm. **(D)** After immunofluorescent staining with anti-HA antibody without permeabilization and imaging by confocal microscopy, the relative amount of surface GLUT4-HA-GFP was quantified by the fluorescent intensity of HA signaling in red divided by the total GLUT4-HA-GFP in green as shown ($n \geq 30$ cells from three independent repeats). **(E–H)** Effect of GSK3 inhibitors on GLUT4-HA-GFP distribution. L6 myoblasts transfected with GLUT4-HA-GFP were treated with DMSO, GSK3α inhibitor (BRD0705, 20 µM), or GSK3β inhibitor (TWS119, 2 µM) under growth (E) or serum-free medium (G) for 3 h. Scale bar, 10 µm. The relative amount of surface GLUT4-HA-GFP was quantified by the fluorescent intensity of HA signaling in red divided by the total GLUT4-HA-GFP in green as shown in F and H ($n \geq 30$ cells from three independent repeats). Data are shown as median ± min and max value. Data are analyzed with one-way ANOVA. *$P < 0.05$; **$P < 0.01$; ***$P < 0.001$.

Dyn2 fission activity, while GSK3β phosphorylation leads to Dyn1 inactivation.

Pharmacological inhibitors of GSK3 have been reported to potentiate insulin action in insulin-resistant rats (Henriksen et al., 2003; Ring et al., 2003). Due to the diversity of their substrates, inhibition of both GSK3 isoforms might be

detrimental as shown by the fatal consequence of GSK3α and GSK3β double deletion in mouse cardiomyocytes (Zhou et al., 2016). There is also a concern regarding β-catenin stabilization, which might arise from dual inhibition of GSK3α and GSK3β (Meijer et al., 2004; Wagner et al., 2016). The improved insulin sensitivity in diet-induced obese mice treated with GSK3α

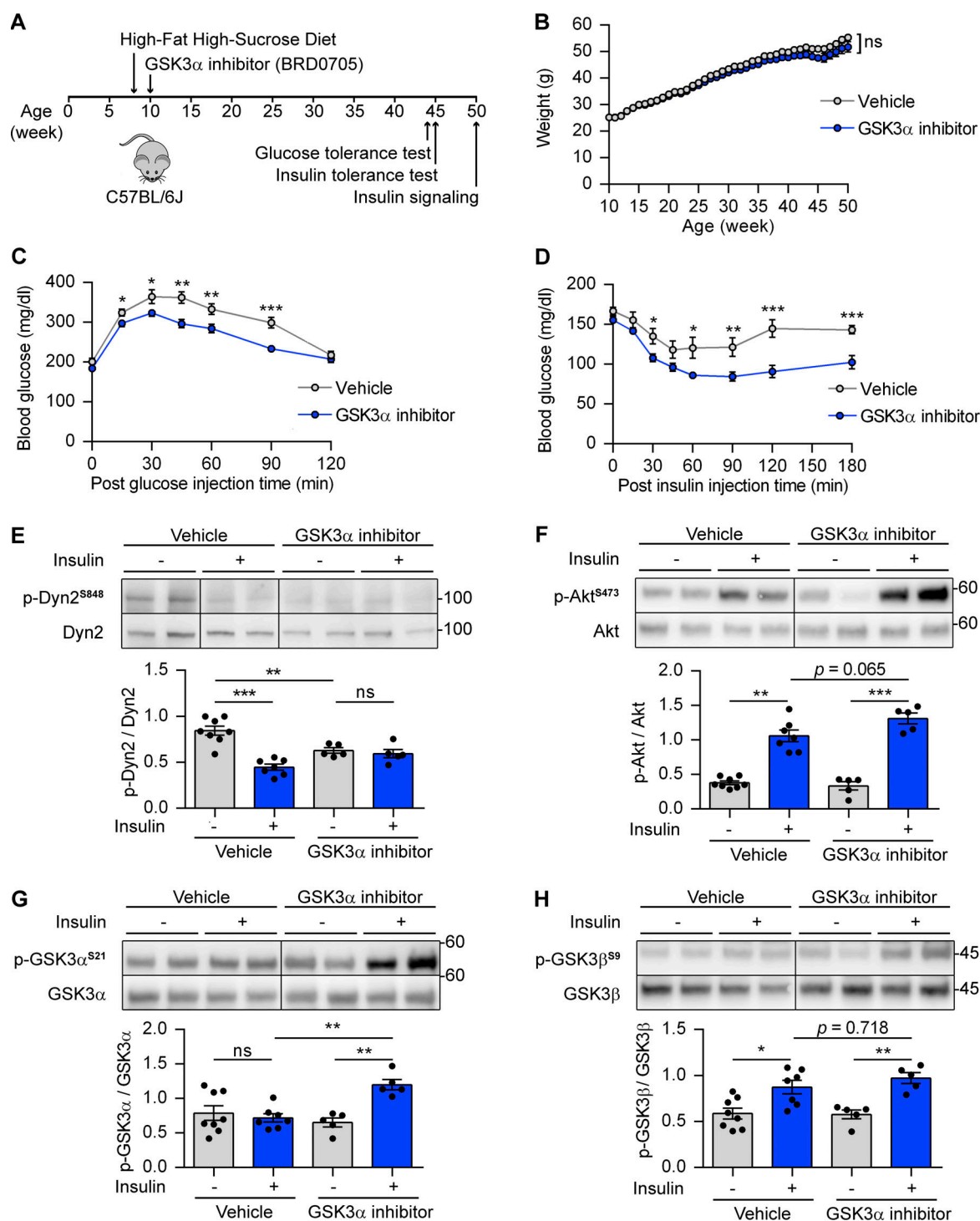

Figure 8. **GSK3α inhibitor improves insulin sensitivity and glucose tolerance in mice fed on HFHSD. (A)** Schematic of GSK3α pharmacological inhibition in mice with HFHSD and the time points for indicated experiments. **(B)** The body weight of mice treated with vehicle or GSK3α inhibitor (BRD0705; $n = 15$ mice per group). **(C and D)** Glucose and insulin tolerance tests after vehicle or BRD0705 were delivered for 34–35 wk. The blood glucose levels were measured at indicated times after peritoneal injection of glucose (C) or insulin (D; $n = 15$ mice for vehicle group, $n = 14$ mice for BRD0705 group). **(E–H)** Effect of BRD0705 on Dyn2 phosphorylation and insulin signaling pathway. Soleus muscle from vehicle- or BRD0705-treated mice were collected 30 min after insulin injection (2 IU/kg), lysed, and detected by Western blotting with indicated antibodies. The ratio of p-Dyn2$^{S848}$ (E), Akt$^{S473}$ (F), GSK3α$^{S21}$ (G), and GSK3β$^{S9}$ (H) normalized to total protein was quantified and shown ($n = 15$ tissue lysates for vehicle group, $n = 10$ tissue lysates for BRD0705 group). Data are shown as average ± SEM and analyzed with Student's $t$ test (two-tailed, unpaired, B–D) or one-way ANOVA (E–H). *$P < 0.05$; **$P < 0.01$; ***$P < 0.001$. Molecular weight is in kD. Source data are available for this figure: SourceData F8.

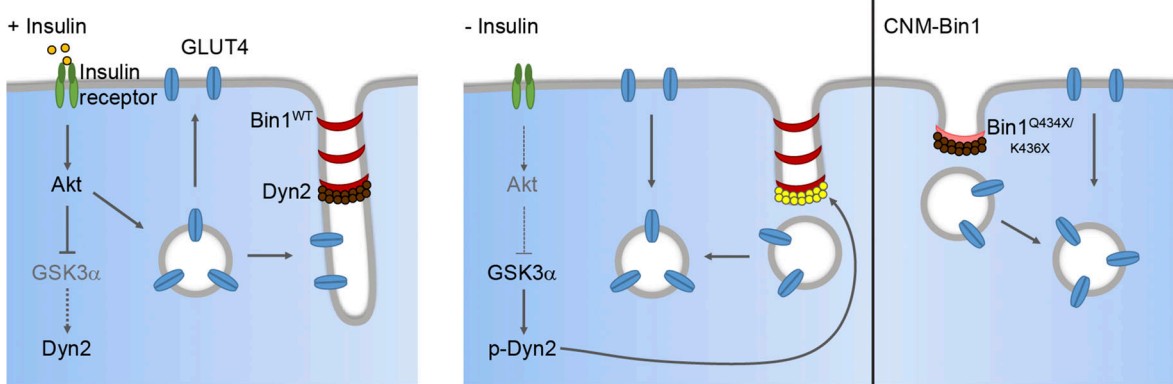

Figure 9. **Dyn2-S848 phosphorylation serves as a molecular switch to promote endocytosis in muscle cells.** Dyn2 activity is inhibited by the binding of Bin1 while GSK3α is inactivated by external signaling, e.g. insulin-PI3K-Akt. The attenuated endocytosis together with the increased exocytosis of GLUT4 lead to its efficient plasma membrane translocation and glucose uptake of muscle upon insulin stimulation. After insulin signaling turns off, GSK3α becomes active and phosphorylates Dyn2-S848 to relieve the Bin1 inhibition and promote endocytosis. The internalization of GLUT4 from muscle surface ceases the glucose uptake of skeletal muscle. CNM-related Bin1 mutations with partial truncation of the SH3 domain lose their inhibitory effect on Dyn2 thus resulting in hyperactive Dyn2 and fragmented T-tubule in muscle cells.

inhibitor from our result, together with its beneficial effects on mouse models of acute myeloid leukemia and fragile X syndrome (McCamphill et al., 2020; Wagner et al., 2018), advocates the therapeutic potential of isoform-specific inhibition of GSK3α in ameliorating metabolic disorders or cancers.

The discovery of insulin in 1921 brought hope and life to millions of people who suffer from diabetes, and the 100-year studies have revealed details of the molecular actions of this amazing hormone in cells and organisms. However, the prevalence of T2D, along with its health and socioeconomic impact, is constantly increasing. T2D is closely associated with many diseases including Alzheimer's disease due to the overlapping cellular and metabolic features shared by these two disorders, including altered glucose homeostasis and insulin resistance (de la Monte and Wands, 2008; Frank and McNay, 2022). Therefore, it is vital to have a better understanding of the molecular events downstream of the insulin. The isoform-specific function of GSK3α we uncovered here would hopefully shed light and bring unique insights into the molecular underpinnings of insulin resistance as well as its therapeutic strategy development.

## Materials and methods
### Protein expression and purification
Human Dyn2 isoform 1 was expressed in Sf9 insect cells transiently transfected with various constructs and purified using GST-tagged SH3 domain of amphiphysin-2. Purified dynamin was dialyzed overnight and stored in a solution containing 20 mM Hepes (pH 7.5), 150 mM KCl, 1 mM DTT, 1 mM EGTA, and 10% glycerol. Various constructs of human Bin1 isoform 8 were cloned into pGEX-4T-1 or pET30a vectors for expression in *Escherichia coli* and purified with glutathione Sepharose beads (GE) or Ni-NTA spin columns (Qiagen), respectively, followed by elution as recommended by the manufacturer. Mouse endophilin A1 was purified like Bin1.

### Preparation of lipid templates
Membrane compositions used in this study were DOPC:DOPS:PI(4,5)P$_2$ at 80:15:5 or DOPC:DOPS:PI(4,5)P2:Rh-PE at 79:15:5:1 for fluorescent-tagged membrane. For liposome preparation, lipid mixtures were dried, rehydrated in buffer containing 20 mM Hepes (pH 7.5) and 150 mM KCl, and then subjected to three rapid freeze–thaw cycles followed by extrusion through a 0.1-μm polycarbonate membrane (Whatman) using Avanti mini extruder. SUPER templates were generated by incubating 2.5-μm silica beads in a solution containing 100 μM fluorescent-tagged liposomes and 1 M NaCl for 30 min at room temperature with intermittent flicking. Excess unbound liposomes were washed four times with filtered water after incubation (Neumann et al., 2013).

### In vitro fission assays
Fission activity of Dyn2 was measured by adding SUPER templates to the top of protein solution containing 0.5 μM Dyn2, 1 mM GTP, and indicated concentrations of Bin1 variants in assay buffer (20 mM Hepes [pH 7.5], 150 mM KCl, and 1 mM MgCl$_2$) for 30 min at room temperature. The mixtures were spun down at a low speed (260 × *g*) in a swinging bucket rotor to separate SUPER template from the released vesicles. Fluorescence intensity was measured in a plate reader (RhPE excitation = 530/25 nm bandwidth and emission = 580/25 nm bandwidth). The fission activity was expressed as the fraction of released lipids from total SUPER templates.

### Liposome binding, TEM, and GTPase activity assays
To analyze membrane binding ability, 0.5 μM Dyn2 and/or 0.5 μM Bin1 were incubated with 150 μM and 100 nm liposomes at 37°C for 10 min. Pellets containing liposome-bound proteins were separated from soluble fractions containing unbound proteins (supernatant) by centrifugation at 15,000 × *g* and resuspended in assay buffer (20 mM Hepes [pH 7.5], 150 mM KCl). Pellet and supernatant were run on SDS-PAGE, stained with Coomassie Blue, and protein intensities were quantified using

ImageJ. Membrane binding ability was expressed as the fraction of membrane-bound proteins divided by total proteins.

To visualize protein assembly on membrane, 1 µM Dyn2 and/or 1 µM Bin1 were incubated with 25 µM liposome for 10 min at 37°C. The mixture was then adsorbed onto the surface of glow-discharged (30 s), carbon film-supported grids (200 mesh copper) for 5 min followed by a 2-min negative-staining with 2% uranyl acetate. Images were captured with a Hitachi H-7650 electron microscope operated at 75 kV and a nominal magnification of 150,000×. Liposome tubule diameter was measured using ImageJ.

For GTPase activity assay, 0.5 µM purified Dyn2 was incubated with 0.5 µM Bin1 or endophilin in a reaction mixture containing 150 µM liposomes, 20 mM Hepes (pH 7.4), 150 mM KCl, 1 µM $MgCl_2$, and 1 mM GTP at 37°C. Liposome-stimulated GTPase activity of Dyn2 was measured as a function of time using a colorimetric malachite green assay, which monitors the release of inorganic phosphate (Leonard et al., 2005).

## GST pulldown assay
To analyze SH3-PRD binding affinity, purified His-Bin1 proteins were first incubated with 100 nm liposomes containing 5% $PI(4,5)P_2$ to alleviate the autoinhibition between SH3 domain and PI motif before being incubated with GST or GST-tagged PRD of Dyn2 immobilized on glutathione beads for 30 min at room temperature. After washing, the beads were boiled in sample buffer (50 mM Tris (pH 6.8), 2% SDS, 10% glycerol, 0.025% bromophenol blue, 1% β-mercaptoethanol), and bound proteins were detected by SDS-PAGE followed by Coomassie Blue staining. ImageJ software was used to quantify protein-band intensity of bound Bin1 relative to GST-PRD. Pulldown assay for purified Dyn2 mutants with GST-tagged SH3 domain of Bin1 or endophilin was performed in a similar manner without the presence of 100 nm liposomes.

## Cell culture, transfection, and lentiviral infection
Mouse-derived C2C12 myoblasts (CRL-1772; ATCC) were cultured in growth medium composed of high glucose (4.5 mg/ml) DMEM supplemented with 2 mM L-glutamine, 1 mM sodium pyruvate, 1× antibiotic-antimycotic, and 10% fetal bovine serum (Gibco). Upon reaching 90% confluency, C2C12 differentiation was induced by replacing growth medium with differentiation medium (DM) composed of high glucose (4.5 mg/ml) DMEM supplemented with 1 mM sodium pyruvate, 1× antibiotic-antimycotic, and 2% horse serum (Gibco). The first time cells were incubated in DM was regarded as day 0 of differentiation. Rat-derived L6 myoblasts (CRL-1458; ATCC) were cultured in low glucose (1.0 mg/ml) DMEM supplemented with antibiotics and 10% fetal bovine serum (Gibco). For transfection, cells at 70% confluency were transfected with a plasmid of interest using Lipofectamine 2000 (Invitrogen) or *Trans*IT-X2 (Mirus Bio) as suggested by the manufacturers. For lentiviral infection, C2C12 myoblasts at 50% confluency were infected with viruses in the presence of 12 µg/ml polybrene followed by puromycin (2 µg/ml) selection 24 h after infection for 3 d before DM (containing 2% FBS) replacement. 10 µg/ml doxycycline was added to day-3-differentiated C2C12, and GLUT4 distribution in cells was

analyzed at day 6 of differentiation. The list of plasmids used for transfection and lentiviral generation in this study can be found in Table S1.

## Fluorescence microscopy
For cells fixed with 4% formaldehyde and permeabilized with 0.1% saponin, images were acquired using a confocal microscope (LSM700, Carl Zeiss) equipped with plan-apochromat 63×/1.35-NA oil-immersion objective (Carl Zeiss) and AxioCam microscope camera, with the excitation laser at 488 or 594 nm and the depletion laser at 592 or 660 nm using Hybrid Detector (Leica HyD) and processed using ZEN software (Carl Zeiss). For time-lapse microscopy, cells were imaged with a spinning-disc confocal microscope or lattice light-sheet microscope (Chen et al., 2014). For the spinning-disc confocal microscope, C2C12 myoblasts were seeded on glass-bottom dishes and cotransfected with Bin1-GFP and Dyn2-mCherry. Cells were placed at 37°C in imaging medium (phenol-red free DMEM with 20 mM Hepes, pH 7.4, 50 µg/ml ascorbic acid, and 10% FBS) and images were captured with 200 ms (488 nm laser) and 400 ms (561 nm laser) exposure and 5-s interval time using Carl Zeiss Cell Observer SD equipped with plan-apochromat 100×/1.40 oil DIC M27 objective (Carl Zeiss) and Photometrics Prime 95B sCMOS camera. For lattice light-sheet microscopy, C2C12 myoblasts cotransfected with Bin1-GFP and Dyn2-mCherry were immersed in an imaging medium (phenol-red free DMEM with 100 U/ml penicillin–streptomycin and 10% FBS) filled chamber at 37°C. Images were illuminated by exciting each plane with a 488 nm laser at 12.56 nano-Watt (nW; at the back aperture of the excitation objective) and 561 nm laser at 37.2 nW, with an excitation outer/inner numerical aperture parameters of 0.55/0.44. Orthogonal to the illumination plane, the water-immersed objective lens (Nikon, CFI Apo LWD 25XW, 1.1 NA, 2 mm WD) mounted on a piezo scanner (PhysikInstrumente, P-726 PIFOC) was used to collect the fluorescence signal, which was then imaged through an emission filter (Semrock Filter: FF01-523/610-25) onto sCMOS camera (Hamamatsu, Orca Flash 4.0 v2 sCOMS) through a 500-mm tube lens (Edmund 49-290, 500 mm FL/50 mm dia; Tube Lens/Tl) to provide the 63× magnification observation. Then the cells were imaged through entire cell volumes at 5-s intervals by using the sample piezo (PhysikInstrumente, P-622 1CD) scanning mode. Finally, raw data were then deskewed and deconvoluted using GPU_decon_bin (Chen et al., 2014), and using Amira software (Thermo Fisher Scientific) to display the 3D images.

## Transferrin uptake assay
For analysis of transferrin internalization, C2C12 myotubes transfected with mCherry-tagged Dyn2 mutants and seeded on fibronectin-coated coverslips were incubated with 5 µg/ml Alexa488-conjugated transferrin for 20 min at 37°C. For recycling efficiency, 30-min incubation with Tfn-488 was followed by 30-min incubation in growth medium without Tfn-488. Cells were washed with ice-cold PBS and then with acid buffer (150 mM NaCl and 150 mM glycine, pH 2.0) repeatedly. Afterward, cells were fixed with 4% formaldehyde and mounted on glass slides using Fluoromount-G mounting medium

(Southern Biotech). Images of the cells were acquired using confocal microscopy and analyzed with ZEN software (Carl Zeiss).

## Subcellular fractionation and analysis of surface:total GLUT4 ratio

For subcellular fractionation, lentiviral infected C2C12 myotubes were homogenized in ice-cold HES-PI buffer (255 mM sucrose, 20 mM Hepes [pH 7.4], 1 mM EDTA, and cOmplete protease inhibitors [Roche]). The lysates were then cleared by centrifugation at 1,000 × $g$ for 5 min at 4°C. After centrifugation at 16,000 × $g$ for 20 min at 4°C, the supernatant was separated from the pellet and subjected to TCA precipitation for 1 h at 4°C. Both pellet and supernatant were resuspended in 2× sample buffer (100 mM Tris [pH 6.8], 4% SDS, 20% glycerol, 0.05% bromophenol blue, 2% β-mercaptoethanol), heated to 60°C for 15 min, and then loaded to 12.5% acrylamide gel for SDS-PAGE and subsequent immunoblotting to detect GLUT4 and markers for heavy membrane ($Na^+/K^+$ ATPase), light membrane (Golgin-97), and cytosol (tubulin). All antibodies used in this study are listed in Table S2. Incubation with HRP-tagged secondary antibody was followed by addition of Immobilon Forte Western HRP substrate (Millipore) to visualize protein signal on membrane.

Insulin-stimulated HA-GLUT4-GFP localization was done according to a previous report (Lampson et al., 2000). Briefly, L6 were seeded on fibronectin-coated coverslips and then cotransfected with Dyn2 mCherry and GLUT4-HA-GFP. Cells were incubated in growth medium or serum-starved for 2 h prior to fixation and subjected to immunofluorescent staining with anti-HA without permeabilization. The cell was imaged under LSM700 confocal microscope (Carl Zeiss) equipped with planapochromat 63×/1.40 oil DIC M27 objective (Carl Zeiss) and analyzed using ZEN software (Carl Zeiss). The relative amount of surface GLUT4-HA-GFP was quantified by the fluorescent intensity of HA signal divided by the total GFP signal.

## Immunoprecipitation

For immunoprecipitation, cells were washed with ice-cold 1x PBS added with 1.5 mM $Na_3VO_4$ and 50 mM NaF before being lysed in lysis buffer (1x PBS, 1% Tx-100, 1.5 mM $Na_3VO_4$, 50 mM NaF, PhosSTOP [Roche] and cOmplete protease inhibitors [Roche]). The lysates were centrifuged at 16,000 × $g$ for 10 min at 4°C. The supernatant was then incubated with anti-HA Agarose (A2095; Sigma-Aldrich, Cat#) for 1 h at 4°C on a rotator. The beads were washed before being boiled in sample buffer for the following SDS-PAGE and immunoblotting. All antibodies used in this study are listed in Table S2. Band intensities were quantified using ImageJ.

## In vitro kinase assay

For in vitro phosphorylation of Dyn2, recombinant Dyn2 was purified from Sf9 as mentioned above, and active recombinant GSK3α was purchased (Cat# 14-492; Millipore). 0.8 µg Dyn2WT or Dyn2S848A was incubated with 10 ng GSK3α in kinase buffer (50 mM Hepes [pH 7.4], 15 mM MgCl2, 200 µM $Na_3VO_4$, 100 µM ATP). The total reaction volume was 30 µl. Following 30 min or 1 h incubation at 30°C, reaction was stopped by adding sample buffer and was boiled at 90°C before being subjected to SDS-PAGE and Western blot analysis with phospho-S848-specific (p-Dyn2$^{S848}$) antibody. p-Dyn2$^{S848}$ antibody is a homemade polyclonal antibody through immunization against phosphoSer848 on Dyn2 using synthetic phosphopeptide with the following sequence: 844 PGVP(pSer)RRPPAAPSRC 858, and purified with affinity column.

## MS

For analysis of Dyn2 phosphorylation status, recombinant Dyn2 phosphorylated by GSK3α in vitro as mentioned above was subjected to SDS-PAGE. The gel was then stained with Coomassie Blue, and bands containing Dyn2 were cut from the gels as closely as possible. The gel pieces were destained, followed by reduction, alkylation, and dehydration before undergoing in-gel digestion with trypsin. The resulting peptides were extracted and went through desalting in C18 column before liquid chromatography with tandem MS (LC-MS/MS) analysis. Peptides were separated on an UltiMate 3000 LCnano system (Thermo Fisher Scientific). Peptide mixtures were loaded onto a 75 µm ID, 25 cm length C18 Acclaim PepMapNanoLC column (Thermo Scientific Scientific) packed with 2 µm particles with a pore of 100 Å. Mobile phase A was 0.1% formic acid in water, and mobile phase B was composed of 100% acetonitrile with 0.1% formic acid. A segmented gradient in 50 min from 2 to 40% solvent B at a flow rate of 300 nl/min. LC-MS/MS analysis was performed on an Orbitrap Fusion LumosTribrid quadrupole mass spectrometer (Thermo Fisher Scientific). Targeted MS analysis was performed with mass accuracy of <5 ppm and a resolution of 120,000 at m/z = 200, AGC target 5e5, maximum injection time of 50 ms followed by high-energy collision-activated dissociation–MS/MS of the focused on m/z 633.8338 (2+) and m/z 422.8916 (3+). High-energy collision-activated dissociation–MS/MS (resolution of 15,000) was used to fragment multiply charged ions within a 1.4 Da isolation window at a normalized collision energy of 32. AGC target 5e4 was set for MS/MS analysis with previously selected ions dynamically excluded for 180 s. The MS/MS spectra of pS848 phosphopeptide were manually identified and checked.

## Mice

All animal experiments were performed according to and approved by the Institutional Animal Care and Use Committee of the Medical College of National Taiwan University (NTU; No. 20201046; IACUC) under standard conditions at 23°C and 12/12 h light/dark (7AM–7PM.) cycle in animal centers of NTU Medical College, which is accredited by the Association for Assessment and Accreditation of Laboratory Animal Care International.

Male C57BL/6J mice are fed on a HFHSD (58% calorie from fat and 12.5% calorie from sucrose, cat no. D12331i, Research Diets) from the age of 8 wk. GSK3α-specific small-molecule inhibitors BRD0705 were dissolved with 3% dimethylacetamide (Sigma-Aldrich) and 10% cremophor (Sigma-Aldrich) in water and were delivered by oral gavage once daily at the dose of 30 mg/kg from the age of 10 wk. Body weight was recorded weekly. At the end of the experiments, mice fasted overnight were i.p. injected with and without human insulin (Humulin-R, Eli Lilly) at the dose of

2 IU/kg. The snap-frozen soleus muscles were harvested 30 min after insulin injection and were homogenized and lysed in radioimmunoprecipitation assay buffer supplied with cOmplete Protease Inhibitor Cocktail (Sigma-Aldrich) and PhosSTOP phosphatase Inhibitor Cocktail (Sigma-Aldrich) for the immunoblotting of total and phospho-Dyn2 and other insulin-signaling components.

### Glucose and insulin tolerance tests
After BRD0705 and the vehicle were delivered for 34–35 wk, i.p. glucose tolerance test was performed in mice fasted for 6 h by i.p. injection of D-glucose water (1 mg/kg). For the insulin tolerance test, mice were fasted for 4 h and injected with human insulin (Humulin-R, Eli Lilly) at the dose of 1.05 IU/kg i.p. Blood was withdrawn from the tail at indicated times after glucose or insulin injection. Blood glucose level was measured using ACCU-CHEK Performa.

### Binding interaction study
Molecular docking analysis was applied to investigate the binding interaction of BRD0705 and GSK3α. The docking analysis was conducted by using the AutoDock4.2 program with the Kollman charge force field (Jiroušková et al., 2009; Morris et al., 2009). Herein, we perform molecular docking simulations to investigate the chemical forces between BRD0705 and GSK3α by structure modeled from PDB 2DFM.

### Statistical analysis
GraphPad Prism 8.0 was used for statistical analysis and the generation of graphs. All data were analyzed with one-way ANOVA followed by Dunnett's or Tukey's post hoc test for comparisons between multiple groups, or Student's $t$ test (two-tailed, unpaired) for comparisons between two groups. Data distribution for $t$ test was assumed to be normal but was not formally tested. Statistical significance was defined using GraphPad Prism 8.0. $P < 0.05$ was considered statistically significant, indicated as $*P < 0.05$; $**P < 0.01$; $***P < 0.001$.

### Online supplemental material
Fig. S1 shows the requirement of PI motif in facilitating membrane binding and tubulation of Bin1, without affecting Dyn2 fission activity. Fig. S2 shows additional characterization of Dyn2-Ser848 phosphorylation in relieving Bin1 inhibition. Fig. S3 specifies the effect of Dyn2-Ser848 phosphorylation in promoting endocytosis, but not exocytosis in cells. Fig. S4 documents the phosphorylation of Dyn2-Ser848 residue by GSK3α. Fig. S5 shows the targeting of Glu196 residue of GSK3α by BRD0705 through molecular docking. Video 1 shows the different dynamics between WT and CNM-Bin1 induced tubules under coexpression of Dyn2 in C2C12 myoblast. Video 2 shows dynamics of Bin1 induced tubules under coexpression with Dyn2-mCherry mutants in C2C12 myoblast. Tables S1 and S2 respectively list plasmids and antibodies used in this study.

### Data availability
All data, plasmids, and reagents used in this study are available upon reasonable request.

## Acknowledgments
We are extremely grateful to Dr. Chun-Liang Pan (NTU) and Dr. Sandra Schmid (Chan Zuckerberg Biohub) for critical reading and helpful comments on this paper. We thank the NTU Consortia of Key Technologies and NTU Instrumentation Center, the imaging core at the First Core Labs, NTU College of Medicine, as well as the electron microscopy core at NTU for their technical support.

This work was supported by National Science and Technology Council grants NSTC 111-2320-B-002-064 and NSTC111-2634-F-002-017 to Y.-W. Liu. We also greatly appreciate grant support from the Collaborative Research Projects of National Taiwan University College of Medicine, National Taiwan University Hospital, and Min-Sheng General Hospital (111C101-81). Open Access funding provided by the National University of Taiwan.

The authors declare no competing financial interests.

Author contributions. All authors participated in the experimental design and conductance; J. Laiman and Y.-W. Liu performed major experiments and analyzed data; W.-C. Tang and B.-C. Chen assisted in live-cell imaging and data interpretation; J. Laiman and Y.-W. Liu wrote the manuscript; B.-C. Chen, L.-M. Chuang, Y.-C. Chang, and Y.-W. Liu supervised the project.

Submitted: 19 February 2021

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

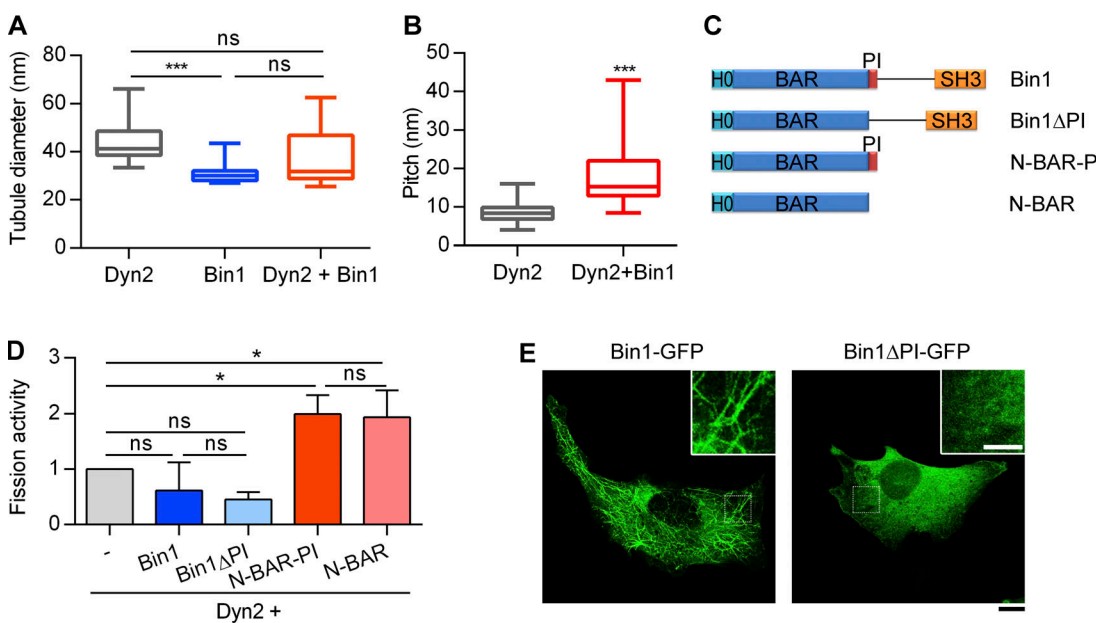

Figure S1. **PI motif of Bin1 is required for membrane binding and tubulation, but not Dyn2 fission activity. (A)** Quantification of the tubulated liposome diameter in Fig. 1 H ($n \geq 11$ liposome tubules). **(B)** Quantification of the periodicity of Dyn2 or Dyn2-Bin1 spirals assembled on liposomes in Fig. 1 H ($n \geq 11$ liposome tubules). **(C)** Domain structure of Bin1 variants used in this study. **(D)** Effect of PI motif from Bin1 on regulating Dyn2 fission activity. 0.5 µM purified Dyn2 was incubated by itself or together with 0.5µM Bin1 variants with or without the PI motif in the presence of GTP and SUPER template. The data was shown as fold change relative to the Dyn2 fission activity alone ($n$ = 3 experimental replicates). **(E)** Effect of PI motif on membrane tubulation ability of Bin1 in cellulo. GFP-tagged Bin1 containing PI motif (Bin1-GFP) or without PI motif (Bin1∆PI-GFP) were over-expressed in C2C12 myoblasts through transfection and viewed under confocal microscopy. Scale bar, 10 µm, scale bar for inset magnification, 5 µm. Bar graphs are shown as average ± SD, box plots are shown as median ± min and max value. Data are analyzed with Student's $t$ test (two-tailed, unpaired; B) or one-way ANOVA (A and D). *P < 0.05; ***P < 0.001.

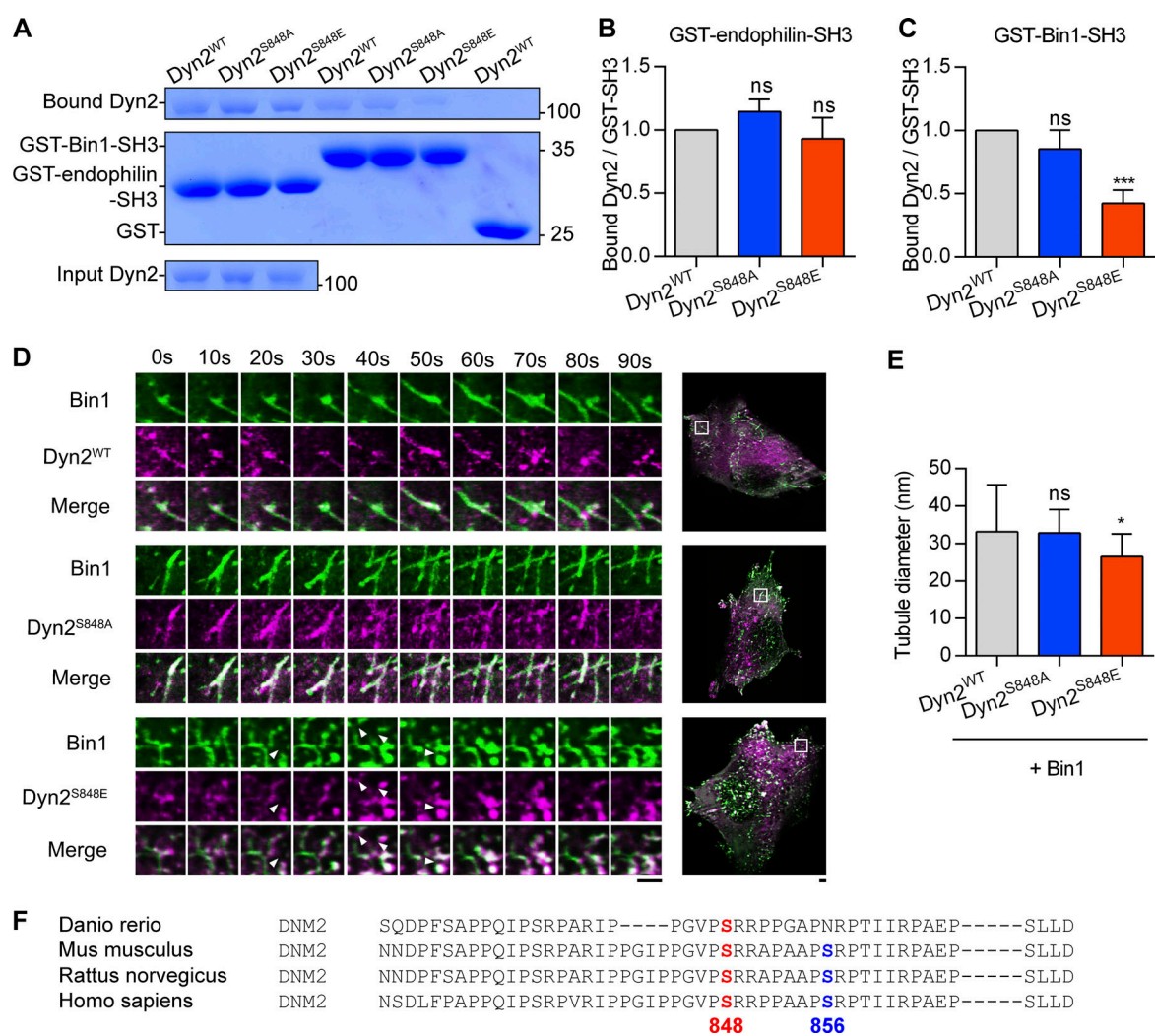

Figure S2. **Dyn2-Ser848 phosphorylation relieves Bin1 inhibition and promotes membrane fission. (A–C)** Binding ability between Dyn2 and different SH3 domain. 12 µg GST, GST-Bin-SH3, or GST-endophilin-SH3 was incubated with 6 µg indicated purified Dyn2. Bound Dyn2 was analyzed with SDS-PAGE and Coomassie Blue staining. The ratio of bound Dyn2 was quantified, normalized to WT, and shown in B and C ($n$ = 3 experimental replicates). Molecular weight is in kD. **(D)** Time-lapse representative images of Bin1-GFP tubules in cells co-expressing WT, S848A, or S848E Dyn2-mCherry were magnified and shown. White arrowheads indicate the occurrence of membrane fission. Scale bar, 2 µm. **(E)** Quantification of tubulated liposome diameter from coincubation of Bin1 and Dyn2 in Fig. 3 G ($n$ ≥ 6 liposome tubules). **(F)** Multiple sequence alignment of Dyn2 PRD domain from different organisms. These sequences were analyzed with EMBL-EBI Clustal Omega Multiple Sequence Alignment. Data are shown as average ± SD and analyzed with one-way ANOVA. *P < 0.05; ***P < 0.001. Source data are available for this figure: SourceData FS2.

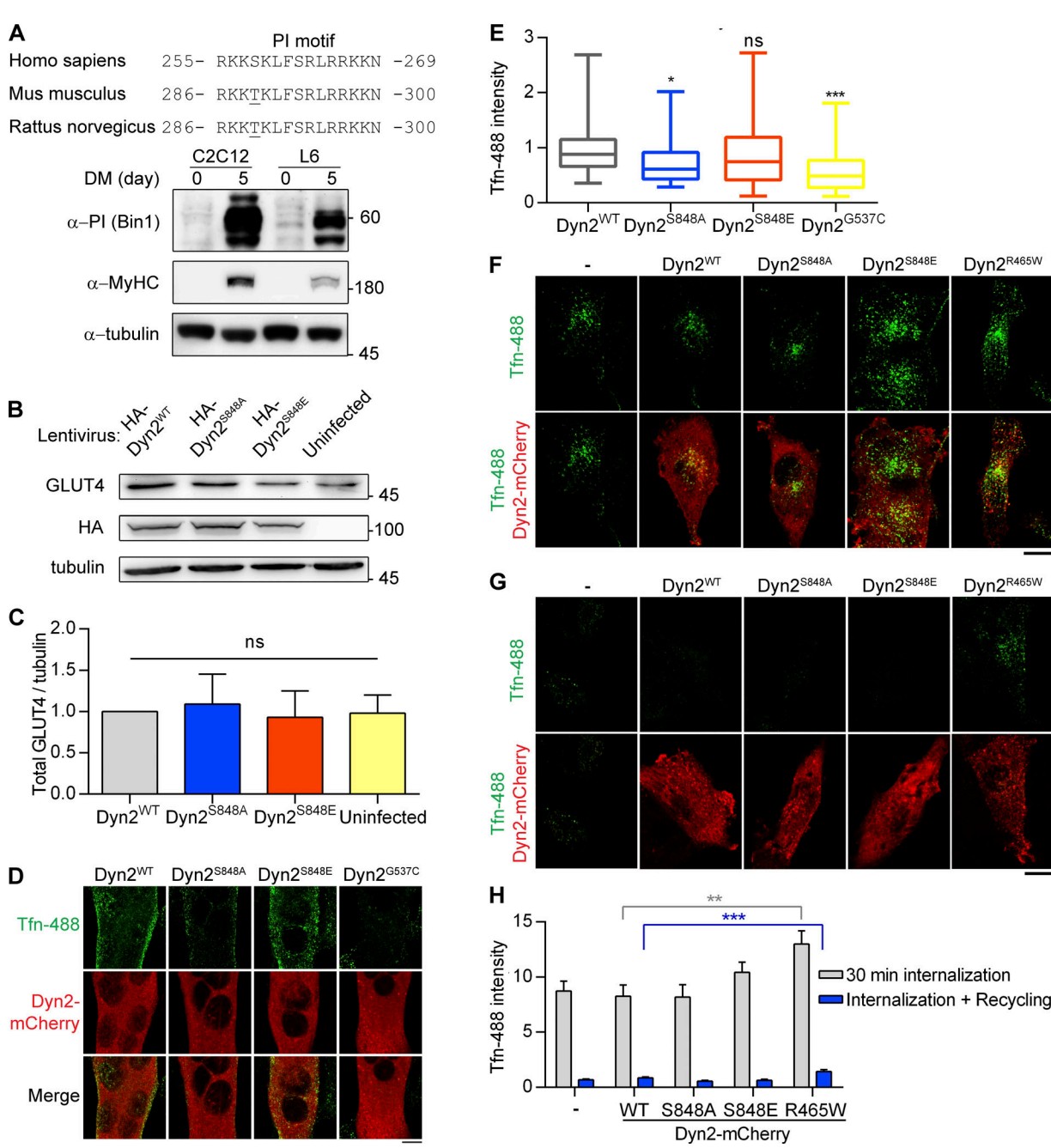

Figure S3. **Dyn2-Ser848 phosphorylation promotes endocytosis, but not exocytosis. (A)** Amino acid sequences of Bin1-PI motif from different organisms (top). The expression of muscle-specific Bin1 in C2C12 or L6 myoblasts cultured in growth medium or DM was examined by Western blotting using anti-PI motif antibody (bottom). **(B and C)** The amount of endogenous GLUT4 in C2C12 myotubes expressing different Dyn2 mutants. C2C12 myotubes infected with indicative HA-Dyn2 mutant lentiviruses expressed a similar level of GLUT4, and the data were quantified, normalized to WT, and shown in B ($n$ = 4 experimental replicates). **(D and E)** Effect of Dyn2 mutants on transferrin internalization in myotubes. C2C12 myoblasts transfected with indicated Dyn2-mCherry mutants and differentiated into myotubes were incubated with Tfn-488 at 37°C for 20 min. Cells were then washed on ice, fixed, and imaged under confocal microscopy. Scale bar, 10 μm. Fluorescence intensity of internalized transferrin was quantified, normalized to Tfn-488 intensity of Dyn2$^{WT}$ transfected cells, and shown in E ($n$ ≥ 24 cells over two independent repeats). **(F–H)** Effects of Dyn2-S848 mutants on transferrin internalization or recycling in L6 myoblast. L6 myoblasts expressing indicative Dyn2-mcherry mutants were subjected to 30-min Tfn-488 uptake, wash, and imaging (F). The recycling efficiency of these internalized transferrin was analyzed by chasing the signal by 30-min incubation of growth medium without Tfn-488 (G). The intensity of internalized Tfn-488 was quantified and shown in H ($n$ ≥ 25 cells over three independent repeats). Scale bar, 10 μm. Bar graphs are shown as average ± SD, box plots are shown as median ± min and max value. Data are analyzed with one-way ANOVA. *$P < 0.05$; **$P < 0.01$; ***$P < 0.001$. Molecular weight is in kD. Source data are available for this figure: SourceData FS3.

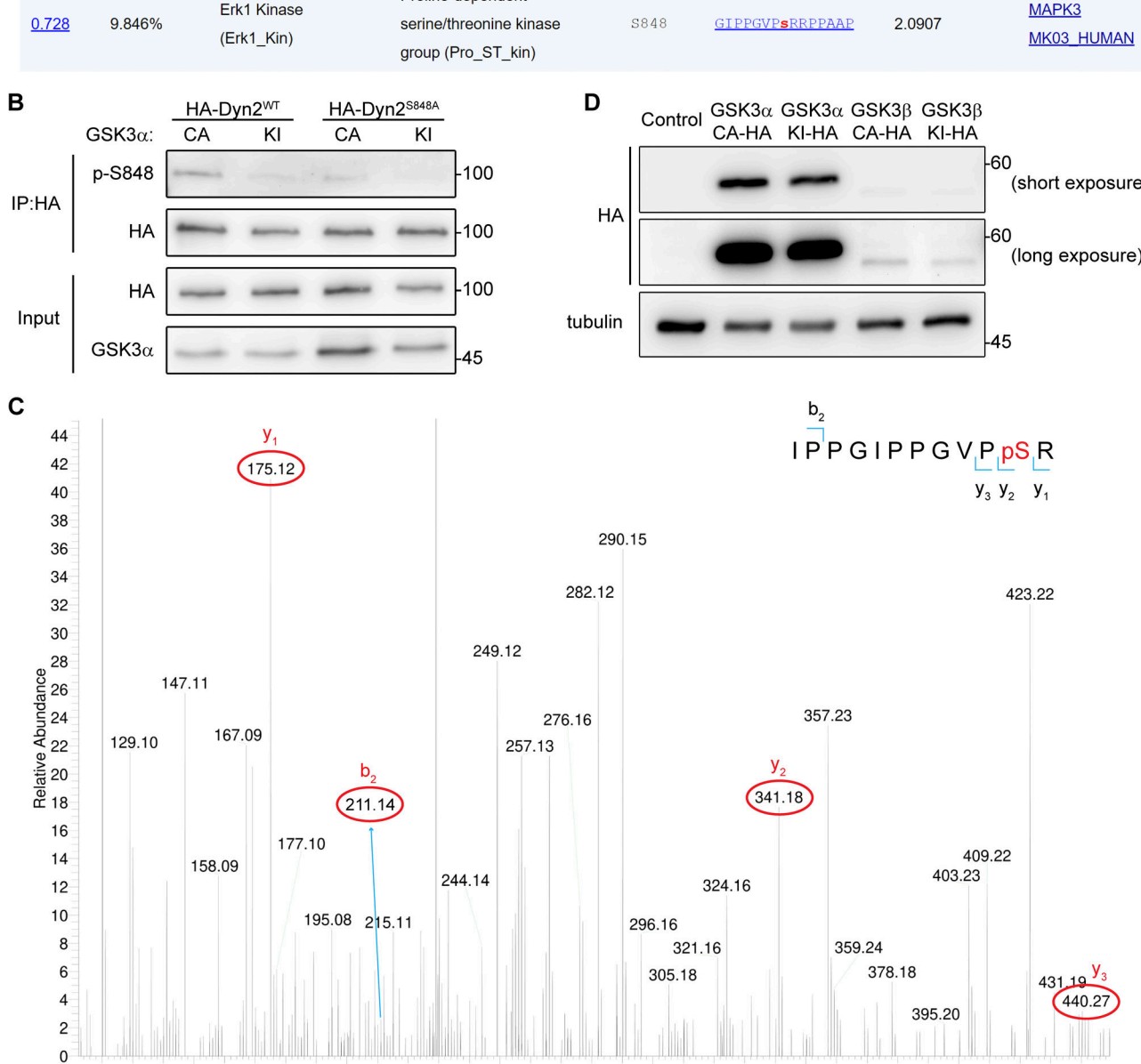

**Figure S4. GSK3α phosphorylates the S848 residue of Dyn2. (A)** Analysis of potential kinases for Dyn2[S848]. Screenshot of the prediction results for Dyn2[S848] kinases using Scansite motif scans (https://scansite4.mit.edu/#scanProtein). Lower score indicates better match of protein sequence to the optimal binding pattern. **(B)** Analysis of Dyn2 phosphorylation under GSK3α overexpression. HA-Dyn2[WT] or HA-Dyn2[S848A] were coexpressed with GSK3α CA or KI in Hela cells. After precipitated with anti-HA antibody, the phosphorylated Dyn2 was detected with Dyn2 phospho-S848-specific (p-S848) antibodies. **(C)** MS analysis of Dyn2 phosphorylation by GSK3α. Recombinant Dyn2 was incubated with GSK3α and ATP in kinase assay buffer for 1 h. The reaction was then analyzed for phosphorylation with LC-MS/MS. The 838-IPPGIPPGVPpSR phosphopeptide containing phosphorylated Ser848 was identified. The MS/MS spectrum of this phosphopeptide is shown. The m/z of (3+) charged phosphopeptide is 422.89 with mass accuracy of <5 ppm. The mass difference between the y1 and y2 ions is consistent with phosphorylation at S848. Of note, the MS/MS spectrum is interfered by another peptide from Dyn2 (258-FFLSHPAYRH) with identical mass to our targeted phosphopeptide containing pS848. **(D)** The expression of HA-tagged GSKα or -β in lentiviral-infected L6 myoblasts was examined by Western blotting using anti-HA antibody. Molecular weight is in kD. Source data are available for this figure: SourceData FS4.

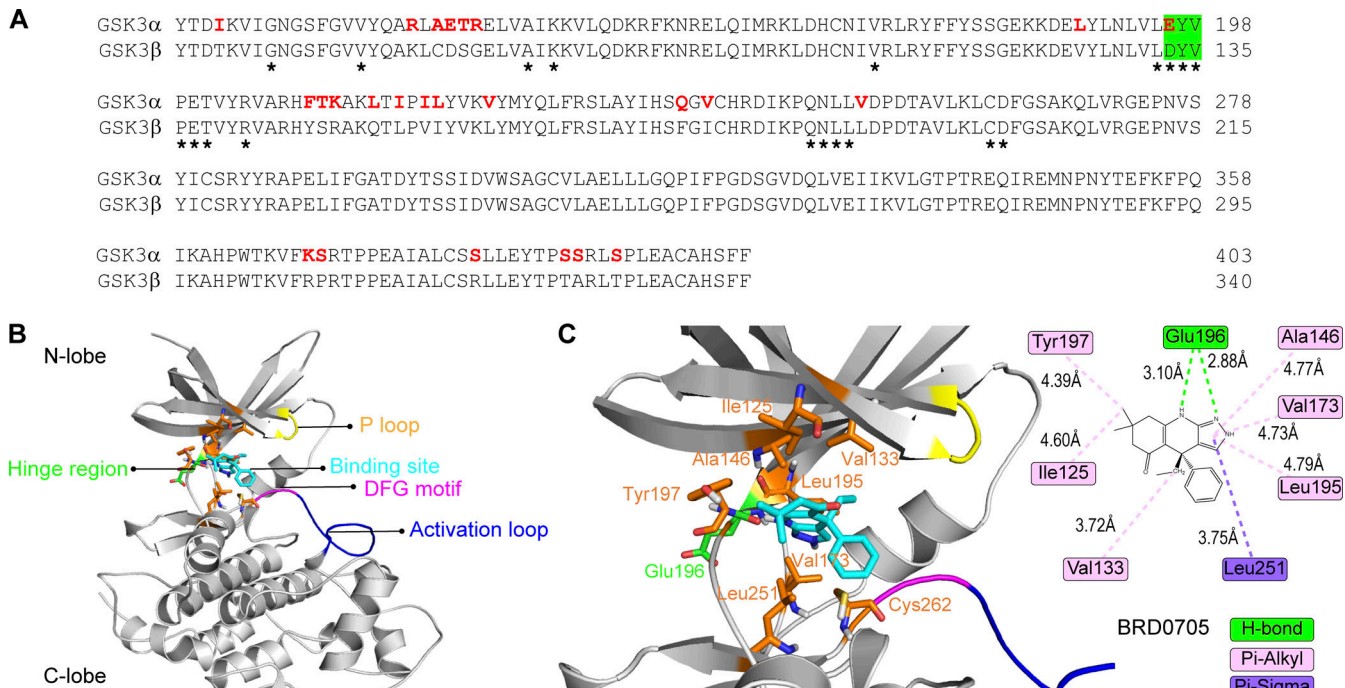

**A**

```
GSK3α  YTDIKVIGNGSFGVVYQARLAETRELVAIKKVLQDKRFKNRELQIMRKLDHCNIVRLRYFFYSSGEKKDELYLNLVLEYV  198
GSK3β  YTDTKVIGNGSFGVVYQAKLCDSGELVAIKKVLQDKRFKNRELQIMRKLDHCNIVRLRYFFYSSGEKKDEVYLNLVLDYV  135
            *        *                  *   *                            *              ****

GSK3α  PETVYRVARHFTKAKLTIPILYVKVYMYQLFRSLAYIHSQGVCHRDIKPQNLLVDPDTAVLKLCDFGSAKQLVRGEPNVS  278
GSK3β  PETVYRVARHYSRAKQTLPVIYVKLYMYQLFRSLAYIHSFGICHRDIKPQNLLLDPDTAVLKLCDFGSAKQLVRGEPNVS  215
       ***  *                                         ****         **

GSK3α  YICSRYYRAPELIFGATDYTSSIDVWSAGCVLAELLLGQPIFPGDSGVDQLVEIIKVLGTPTREQIREMNPNYTEFKFPQ  358
GSK3β  YICSRYYRAPELIFGATDYTSSIDVWSAGCVLAELLLGQPIFPGDSGVDQLVEIIKVLGTPTREQIREMNPNYTEFKFPQ  295

GSK3α  IKAHPWTKVFKSRTPPEAIALCSSLLEYTPSSRLSPLEACAHSFF                                     403
GSK3β  IKAHPWTKVFRPRTPPEAIALCSRLLEYTPTARLTPLEACAHSFF                                     340
```

Figure S5.   **BRD0705 targets the Glu196 residue of GSK3α. (A)** Sequence alignment of the kinase domain of GSK3α and GSK3β. The hinge region was highlighted in green. Residues within 6 Å approximity of the ligands are marked with asterisk. **(B and C)** Molecular docking of GSK3α and BRD0705. Interaction between amino acid side chain and the inhibitor was annotated in C.

Video 1.   **Time-lapse microscopy of Dyn2$^{WT}$-mCherry and Bin1-GFP mutants in C2C12 myoblasts (related to Fig. 2 L).** C2C12 myoblasts cotransfected with Dyn2$^{WT}$-mCherry (magenta) and Bin-GFP mutants (green) were imaged with a spinning-disc confocal microscope with 5-s interval. The video shows the different dynamics between WT and CNM-Bin1 induced tubules under coexpression of Dyn2. The video plays at 30 frames per second.

Video 2.   **Time-lapse microscopy of Dyn2-mCherry mutants and Bin1$^{WT}$-GFP in C2C12 myoblasts (related to Fig. S2 D).** C2C12 myoblasts cotransfected with Dyn2-mCherry mutants (magenta) and Bin1$^{WT}$-GFP (green) were imaged with lattice light-sheet microscopy with 5-s interval. The video plays at 30 frames per second.

**Provided online are two tables. Table S1 lists plasmids used in this study. Table S2 lists antibodies used in this study.**

