## [Peer Review File · The Journal of Cell Biology]

GSK3 α phosphorylates dynamin-2 to promote GLUT4 endocytosis in muscle cells

Jessica Laiman, Yen-Jung Hsu, Julie Loh, Wei-Chun Tang, Mei-Chun Chuang, Hui-Kang Liu, Wei-Shun Yang, Bi-Chang Chen, Lee-Ming Chuang, Yi-Cheng Chang, and Ya-Wen Liu

Corresponding Author(s): Ya-Wen Liu, National Taiwan University and Yi-Cheng Chang, National Taiwan University Hospital

Review Timeline:

Submission Date:	2021-02-19
Editorial Decision:	2021-05-10
Revision Received:	2022-08-17
Editorial Decision:	2022-10-12
Revision Received:	2022-10-18

Monitoring Editor: Satyajit Mayor

Scientific Editor: Lucia Morgado-Palacin

Transaction Report:

DOI: <https://doi.org/10.1083/jcb.202102119>

May 10, 2021

Re: JCB manuscript #202102119

Prof. Ya-Wen Liu
National Taiwan University
Institute of Molecular Medicine
No. 1, Sec. 1, Jen-Ai Rd., R1517
Taipei 10002
Taiwan

Dear Prof. Liu,

Thank you for submitting your manuscript entitled "Muscle-Specific GLUT4 Endocytosis Regulated by GSK3a-Dyn2-Bin1 Interplay". The manuscript has been evaluated by expert reviewers, whose reports are appended below. Thank you very much for your patience while we were assessing the review reports. Unfortunately, after an assessment of the reviewer feedback, our editorial decision is against publication in JCB.

As you will see, while the reviewers have voiced some enthusiasm for the premise of your work, they have also raised a number of significant concerns. In particular, they feel that some of the main conclusions are not adequately supported by the data and a more rigorous interrogation would be needed in order for the paper to be deemed suitable for JCB (see reviewers #1 and #2 specific comments on this latter point). In addition, the reviewers feel that further evidence is needed to demonstrate the physiological/functional relevance of your conclusions.

Based on the extent of revisions that would be necessary to address the reviewers' concerns, we cannot consider your manuscript for publication in JCB at this time. If you wish to expedite publication of the current data, it may be best to pursue publication at another journal.

However, given interest in the topic, we would be open to an appeal of this decision and resubmission to JCB of a significantly revised and extended manuscript that completely addresses each of the reviewers' concerns in full. If you choose to follow this path, we encourage you to discuss with the journal office the suitability of a revision plan before resubmitting. Once we have discussed such revision plan, your revised manuscript will be treated as an appeal, which should be submitted directly through our manuscript submission system. Please note that priority and novelty would be reassessed at resubmission and the paper would, of course, be subject to re-review by the same reviewers (if possible).

Regardless of how you choose to proceed, we hope that the comments below will prove constructive as your work progresses. You can contact the journal office with any questions, cellbio@rockefeller.edu.

Thank you for thinking of JCB as an appropriate place to publish your work.

Sincerely,

Satyajit Mayor
Monitoring Editor
Journal of Cell Biology

Lucia Morgado Palacin, PhD
Scientific Editor
Journal of Cell Biology

Reviewer #1 (Comments to the Authors (Required)):

Centronuclear myopathy (CNM) which is associated with mutations in MTM1, DNM2 and BIN1, resulting in hyperactive Dyn2 GTPase and endocytic vesicle fission. This study analyses the regulation of Dyn2 fission activity via: a) its interactor Bin1 and b) phosphorylation of Ser848 near a proline-rich site that interacts with the SH3 domain of Bin1.

Bin1 and Dyn2 interact on membranes and Bin1 favours tubule formation and suppresses Dyn2 fission activity. Bin1deltaSH3 or disease associated mutants of Bin1 prevent this negative regulation. In C2C12 muscle cells Bin1-GFP and Dyn2-mCherry

localized to tubules, but Bin1 mutants were less able to do so. Moreover, Dyn2 S848E bound less effectively to GST-Bin1-SH3 than Dyn2-S856E. Dyn2-S848E-mCherry promoted more vesicle formation than Dyn2(WT)-mCherry in C2C12 myotubes, suggesting less negative regulation of Dyn2 fission activity. Dyn2-S484A had strong negative impact on Dyn2 fission activity versus high fission activity of Dyn2-S848E. These results suggest that Dyn2 S848 phosphorylation lessens its interaction with and regulation by Bin1.

Scansite 4.0 analysis yielded GSK3alpha as a kinase candidate for S848 on Dyn2. Constitutively active GSK3alpha phosphorylated HA-Dyn2, but not Dyn2-S848A, in L6 myoblasts. Serum starvation increased pSer on Dyn2WT, but not on Dyn2-S848A and increased GSK3. A phospho-specific antibody for Dyn2-S848 was generated that detected GSK3alpha phosphorylation of Dyn2WT in vitro.

To investigate the consequence of such phosphorylation, the authors studied transferrin uptake in C2C12 myotubes. Transferrin-A488 uptake was reduced by Dyn2-S848A, which was considered above to be tonically inhibited by Bin1 binding. However, Dyn2-S848E did not increase Tfn-A488 uptake. The study then switches to analyze the effect of Dyn on GLUT4 localization in L6 muscle cells, because GLUT4 is known to undergo endocytosis requiring Dyn. As is also well known, GLUT4 mobilizes to the plasma membrane in response to serum or insulin, and here the authors show that DynS848A tended to retain GLUT4-GFP intracellularly.

Major comments

1. This study consists of two parts that are not properly connected. The first part analyzes in greater detail the regulation of Dyn2 by Bin1 and by phosphorylation, and makes a case for how this impacts on Transferrin uptake. The uptake needs to be better quantified as endocytic and exocytic rates, but otherwise it is a fine piece of work detailing the regulation of Dyn2. The second part, focussing on GLUT4, is disconnected and not analyzed deeply enough. For this study to be more cohesive, the entire section on GLUT4 must either be substantially beefed up or removed from this manuscript, as presented. The fact that Dyn2 regulates transferrin already shows that any action on other proteins that require dynamin for endocytosis may be equally affected, so that effects on GLUT4 would not be surprising in any case.
2. The conclusion that our findings reveal a new regulatory mechanism for Dyn2 activity in muscle cell that plays an important role in glucose metabolism and muscle physiology is totally unfounded as none of this is investigated in this study. Even if the GLUT4 traffic parameters were properly assessed, the above statement is an exaggeration when based solely on cell culture studies.
3. The Introduction ends proposing a role of Dyn2 at the muscle T-tubules but the study is all performed in cell cultures that do not form T-tubules. Related to this, what is the relationship between the intracellular tubes shown in cells and T-tubules in a mature muscle?
4. The subcellular fractionation and characterization suggest the P fraction has plasma membranes, since it expresses the Na/K-ATPase, but it also has a substantial amount of EEA1 contamination which is an early endosome marker. In addition, it is essential to show if any of the fractions contain TGN or recycling endosome markers or a GLUT4 vesicle marker such as VAMP2. More analysis and discussion of these fractionation results is warranted to support the conclusion that Dyn2-S848A is elevated at the PM relative to Dyn2-S848E.
5. Related to the above, the effect of Dyn mutants on total GLUT4 levels must be shown, as mislocalization of GLUT4 can result in its degradation at the lysosome.
6. The assignment of GLUT4-GFP puncta to the plasma membrane versus intracellular vesicles is not rigorously demonstrated. Typically, a fluorescent marker of the plasma membrane would be co-expressed with GLUT4-GFP or there are GLUT4 chimeric proteins with extracellular epitopes of HA or Myc that have been created for this purpose. This reviewer would insist that Figure 5B to G be repeated with a GLUT4-7myc-GFP construct that can be obtained from Addgene.
7. Moreover, to be able to conclude that Dyn2 phosphorylation regulates GLUT4 endocytosis the authors should use the above construct to directly measure GLUT4 endocytosis, which would greatly solidify the conclusions of the effect of Dyn2WT and its mutant S848A and S848E.
8. What was the rationale for using C2C12 muscle cells for Transferrin uptake and L6 for GLUT4-GFP distribution? And why myotubes of the former but myoblasts of the latter?
9. It would be essential to test if serum or insulin regulate Dyn2 phosphorylation in muscle cells. Related to this, what is the effect of inhibiting GSK3alpha on transferrin uptake (or GLUT4 localization)?
10. Does the viral vector result in equivalent expression of GLUT4-GFP in all cells? Otherwise, it would be important to normalize the membrane and intracellular localization of GLUT4 to the expression in each cell. Related to this, myoblasts are rather thin cells, so that assigning surface to intracellular localization would highly benefit from showing confocal planes of

analysis.

Minor issues

1. The authors state that C2C12 myotubes express more Bin1 than C2C12 myoblasts. It would be of interest to illustrate this with immunoblotting or qPCR. Along these lines, do L6 myoblasts express Bin1, could L6 myotubes express more Bin1, do L6 myoblasts express more Bin1 than C2C12 myoblasts on a per protein basis.
2. It is stated that the serum condition is physiological, but 5% serum is not precisely so.
3. Does the Dyn2 phospho-S848 antibody detect endogenous Dyn2 in L6? If so, some of the above experiments should be performed with the endogenous Dyn 2 (such as testing its phosphorylation status with serum or insulin).
4. Scansite 4.0 was used to discover GSK3alpha as a Dyn2-Ser848 kinase. It would be of interest if the authors showed or described this analysis (including a citation of website hyperlink) in supplemental figures to demonstrate how highly ranked GSK3alpha appeared and if other kinases were also ranked highly.

Reviewer #2 (Comments to the Authors (Required)):

Laiman et al. present very interesting insights into dynamin functions and how these could be regulated in response to insulin signaling in the context of muscle cells. Using a combination of biochemistry and live cell imaging, the authors show that interactions between BIN1's SH3 domain and dynamin2's PRD inhibits dynamin2's capacity for membrane fission, explaining why BIN1-coated tubules in cells are resilient to severing by dynamin. Furthermore, the authors address reasons why certain CNM-specific mutants of BIN1 appear to be organized not as tubules but as vesicles in cells. These CNM-specific mutants, which result in a partial truncation of its SH3 domain, appear to weaken interactions with dynamin2 and consequently get severed by dynamin. The authors then suggest a physiological signaling axis managed by insulin dependent GSK3alpha that phosphorylates dynamin at its PRD on S848, which weakens its ability to interact with BIN1 and facilitates scission of BIN1 tubules. The authors extend these results to understand how such a kinase-controlled partner interaction influences dynamin's functions during CME in muscle cells using GLUT4 as a marker cargo. The same insulin-dependent signaling apparently contributes to facilitating GLUT4 endocytosis via CME.

The results described in the manuscript can be divided into two parts, which are not necessarily linked. The first addresses BIN1-dynamin interactions while the second address dynamin's role in CME. The experiments carried out for the first part are well-designed, meticulous and convey novel information on dynamin's function with respect to the stability of BIN1 tubules. The data presented on the mechanism for why the CNM-specific mutants of BIN1 are quite significant and could have a profound impact on understanding the pathology of CNM. Regrettably, I find the second part lacking a clear rationale and could benefit from more pointed experiments and rigorous analyses.

While this work presents novel insights into why certain CNM-associated BIN1 mutants display an altered vesicular morphology, I was unsure of the exact mechanism for these effects. The mutations cause a partial truncation of the SH3 domain in BIN1 but surprisingly, they seem to fare as well as an SH3 deletion construct of BIN1 in fission assays. How do the authors explain this? The authors state that a relatively small decline in binding affinity between SH3-PRD interactions is sufficient to rescue dynamin function but it would be useful to estimate the binding affinities. In other words, how much of a lowering in binding affinity would be necessary to strike a balance between efficient recruitment of dynamin to BIN1-coated tubules without causing an inhibition in its fission activity?

The suggested model that insulin-dependent signaling regulates dynamin functions is indeed interesting. In the context of BIN1, this link is largely inferred from the use of phosphomimetic mutants of dynamin2, which is understandable for biochemical experiments. But establishing that such signaling indeed causes a partial disengagement of dynamin from BIN1 tubules in cells and a subsequent severing of the tubule could provide strong validation of this model. This can be easily assayed by monitoring the effect of an acute application of insulin to cells followed by the two-color imaging of BIN1 and dynamin. Was this checked?

The in vitro data on fission manifests from dynamin's ability to vesiculate the planar SUPER templates. Quite likely, such vesiculation relies on an upstream process that causes tubules to grow from the template which are then captured and severed by dynamin. Dynamin1's strong tendency to self-assembly causes both tubulation and severing of the membrane. But dynamin2 requires other tubulating proteins such as endophilin and BIN1. How well does this assay recapitulate cellular physiology? Is the growth of BIN1 tubules in cells influenced by the presence of endogenous dynamin? Clearly BIN1 inhibits dynamin-catalyzed vesiculation from SUPER templates but does a complex of BIN1 and dynamin even tubulate SUPER templates? Some of these points could be clarified in the manuscript.

The part about insulin-dependent regulation of dynamin function in GLUT4 trafficking appears quite preliminary and the suggested model is tenuous at best. Based on how the manuscript is written, I'm not sure if the authors are trying to emphasize

a common link between dynamin's function in BIN1 tubule dynamics and in CME. If yes, then isoform 8 of BIN1 does not have any clathrin and AP2 binding sites so it's unlikely to even be recruited to CCPs. Such a link therefore lacks justification at the level of participant proteins. The effects of insulin signaling on CME should therefore manifest from a regulation of dynamin's interaction with some other binding partner. But as the authors themselves convince us so nicely of the fact that not all SH3 interactions are inhibitory to dynamin's function in fission, even a causal link appears vague at best.

In sum, I think the manuscript has some very nice data, especially those that pertain to the CNM-specific mutants of BIN1. I believe these would generate quite a bit of excitement in the community. I would recommend a shorter but more focused manuscript, emphasizing the BIN1-dynamin interactions addressing the experiments and clarifications to text suggested above.

Reviewer #3 (Comments to the Authors (Required)):

Laiman and colleagues propose experiments and data showing the interplay between Bin1, Dyn2 and GSK3 in vitro and in cells, and the impact on membrane tubulation and on GLUT4 endocytosis. As Bin1 and Dnm2 are mutated in a human disease, they also convincingly decipher the impact of some of the mutations. They discovered a regulation of Dyn2 by phosphorylation through GSK3, and a 'bi-directional' regulation by Bin1 where the N-BAR domain of Bin1 favors Dyn2 membrane fission while the SH3 domain inhibits Dyn2-mediated fission.

Several experiments revealed such mechanism appears specific to Bin1, at least not conserved with endophilin, which is very interesting. Similarly the experimental and literature data comparing the regulation of Dyn1 and Dyn2, especially through different GSK3 isoforms, are exciting and should most probably be of interest for future studies.

The concept of Bin1 negatively regulating Dyn2 was shown before in vitro and in vivo including in mammals, as the effect of the SH3 mutations on Dyn2 binding, and is thus not brand new. However, as the authors pointed, they made a great job to decipher the underlying molecular mechanism concerning this part. The regulation of GLUT4 trafficking through GSK3-mediated Dyn2 848 phosphorylation appears entirely novel.

Major points:

-phosphorylation status of Dyn2 WT in several experiments:

fig4D: it is interesting but unclear to me how/why S848A leads to a Bin1-mediated inhibition of Dyn2 fission activity. In Fig4C there seems to be an increase in Bin1 binding (compared to Dyn2 WT) that would fit but how this is happening? Probably the authors have envisaged Dyn2 produced from sf9 cells is partially phosphorylated: they can test that with their antibody. Also in the other experiments, did they use phosphatases or inhibitors in their Dyn2 preparation to be sure they were performed with a Dyn2-WT with known phosphorylation status?

Similarly, in transfected cells, is the transfected Dyn2 WT partially phosphorylated; that sounds feasible as the S848A fissions less.

Similarly in Fig5 GLUT4 ratio, Dyn2 WT seems to have intermediate values between the non-phosphorylated 848A and the pseudo-phosphorylated 848E, again suggesting a partial phosphorylation that could have been demonstrated or ruled out with inhibitors and their anti-phospho antibodies (either p-S848-Dyn2 or as in Fig6A).

-Concerning the experiments testing the role of the PI domain and of mutations truncating the SH3, how do they impact on Bin1 conformation and does this impact on the regulation of Dyn2? Indeed Kojima et al 2004 suggested the PI domain binds the SH3 domain of Bin1. How do the authors take into account or test that the deletion of the PI or SH3 domains unmasks one of these domains? This Bin1 autoinhibition appears an important mechanism that might alter the conclusion of the experiments investigating the regulation of Dyn2 functions.

On a similar line, they found the deletion of the PI domain, supposedly 'opening' Bin1, has no effect on Dyn2 fission activity. Can they study this point with pulldown as in figs 2 and 4 (is PI domain altering Dyn2 binding through itself or the SH3).

Minor points:

-is there a way to quantify results of figure 1F, as the helix organization seems to be rather preserved with Bin1+ Dyn2, to definitely conclude Bin1 alters the assembly of membrane-bound Dyn2 (in terms of spiral pattern, not on tubules diameter that is reported in figure 1G).

-is there a statistical increase in Dyn2 fission with N-BAR-PI compared to Bin1 in fig S2B as in Fig 1? or none of the conditions are statistically different?

-FigS2H: it will be better to compare +/-PI domain with the same tag as the use of different tags may impact on potential dimerization; to rule out BIN1-GFP tubulates due to GFP and not because of the addition of the PI domain.

-FigS3C: tubules look thicker with the mutants in cells; any confirmation of that, or was it similar on liposomes for Bin1 alone?

-Fig3C: in the time lapse, if possible it will be more informative to show Dyn2-mCherry together with Bin1-GFP, to assess if fission is happening upon recruitment of Dyn2.

-Fig4 and associated data: the rationale to select the 848 and 856 sites may be clarified as the present findings seem to be based on the description of the 848 phosphorylation in Efendiev 2002. However there are other PXXP motifs nearby that were shown to bind to amphiphysin but were not tested (eg QIPSRPVR). Do the authors think or have shown that phosphorylation at

these other sites is not implicated in Bin1 binding and regulation.

-Fig5C-D: the way to quantify surface GLUT4 with the intensity ratio surface/total is assuming only dot-like pattern of GLUT4-GFP is intracellular; subcellular fractionation as in 5A might probably be more precise.

-top of p19: may precise 'without affecting the extent of phosphorylation of Dyn2-848 MUTANT' (or 848A)

-Discussion: The authors conclude that disruption of t-tubule homeostasis is the common hallmark of CNM. They also showed impact on CME and GLUT4; could they speculate on the impact of these defects in the disease.

-Please clarify in methods which Dyn2 splice isoform was used as the Schmid's team and others showed they have different functions

-In general it would help to indicate the number of biological replicates in each legends.

Point-by-point response to reviewers:

JCB manuscript #202102119/

"Muscle-Specific GLUT4 Endocytosis Regulated by GSK3 α -Dyn2-Bin1 Interplay"
(revised title: GSK3 α , but not GSK3 β , phosphorylates dynamin-2 to promote GLUT4 endocytosis in muscle cells.)

Reviewer #1 (Comments to the Authors (Required)):

Centronuclear myopathy (CNM) which is associated with mutations in MTM1, DNMT2 and BIN1, resulting in hyperactive Dyn2 GTPase and endocytic vesicle fission. This study analyses the regulation of Dyn2 fission activity via: a) its interactor Bin1 and b) phosphorylation of Ser848 near a proline-rich site that interacts with the SH3 domain of Bin1.

Bin1 and Dyn2 interact on membranes and Bin1 favours tubule formation and suppresses Dyn2 fission activity. Bin1 Δ SH3 or disease associated mutants of Bin1 prevent this negative regulation. In C2C12 muscle cells Bin1-GFP and Dyn2-mCherry localized to tubules, but Bin1 mutants were less able to do so. Moreover, Dyn2 S848E bound less effectively to GST-Bin1-SH3 than Dyn2-S856E. Dyn2-S848E-mCherry promoted more vesicle formation than Dyn2(WT)-mCherry in C2C12 myotubes, suggesting less negative regulation of Dyn2 fission activity. Dyn2-S848A had strong negative impact on Dyn2 fission activity versus high fission activity of Dyn2-S848E. These results suggest that Dyn2 S848 phosphorylation lessens its interaction with and regulation by Bin1.

Scansite 4.0 analysis yielded GSK3 α as a kinase candidate for S848 on Dyn2. Constitutively active GSK3 α phosphorylated HA-Dyn2, but not Dyn2-S848A, in L6 myoblasts. Serum starvation increased pSer on Dyn2WT, but not on Dyn2-S848A and increased GSK3. A phospho-specific antibody for Dyn2-S848 was generated that detected GSK3 α phosphorylation of Dyn2WT in vitro.

To investigate the consequence of such phosphorylation, the authors studied transferrin uptake in C2C12 myotubes. Transferrin-A488 uptake was reduced by Dyn2-S848A, which was considered above to be tonically inhibited by Bin1 binding. However, Dyn2-S848E did not increase Tfn-A488 uptake. The study then switches to analyze the effect of Dyn on GLUT4 localization in L6 muscle cells, because GLUT4 is known to undergo endocytosis requiring Dyn. As is also well known, GLUT4 mobilizes to the plasma membrane in response to serum or insulin, and here the authors show that DynS848A

tended to retain GLUT4-GFP intracellularly.

>> We are grateful for these helpful comments from the reviewer. We have hence revised our manuscript accordingly, and the specific responses to reviewer comments are listed below.

Major comments

1. This study consists of two parts that are not properly connected. The first part analyzes in greater detail the regulation of Dyn2 by Bin1 and by phosphorylation, and makes a case for how this impacts on Transferrin uptake. The uptake needs to be better quantified as endocytic and exocytic rates, but otherwise it is a fine piece of work detailing the regulation of Dyn2. The second part, focussing on GLUT4, is disconnected and not analyzed deeply enough. For this study to be more cohesive, the entire section on GLUT4 must either be substantially beefed up or removed from this manuscript, as presented. The fact that Dyn2 regulates transferrin already shows that any action on other proteins that require dynamin for endocytosis may be equally affected, so that effects on GLUT4 would not be surprising in any case.

>> We thank the reviewer for the constructive suggestions. We thus have beefed up the GLUT4 section by quantifying the surface/total GLUT4 ratio, both the endogenous and exogenously expressed, in C2C12 and L6 myoblasts (Fig 4). Most importantly, we have restructured the manuscript to focus on the insulin signaling-dependent regulation of Dyn2 activity mediated by Bin1 and GSK3 α .

It has been well-studied that the exocytosis of GLUT4 is promoted by insulin signaling. In this work, we find that the internalization of GLUT4 is also regulated by insulin signaling via GSK3 α -mediated Dyn2 inactivation that underpins the improved glucose tolerance and insulin sensitivity in type 2 diabetic mice treated with GSK3 α inhibitor (Fig 8). Therefore, we focus on GLUT4 endocytosis and use transferrin uptake merely as supporting data (Fig S3 D, E)

To address the effects of Dyn2 mutations on exocytosis, we examined the efficiency of Tfn-488 recycling and observed no significant difference in myoblasts expressing different Dyn2-S848 mutants (Fig S3F-H). Together with the similar surface ratio of GLUT4-HA-GFP in L6 myoblasts expressing Dyn2^{S848A} and Dyn2^{WT} cultured in growth medium (Fig 4C and E), these results suggest that the lower endocytic rate of Dyn2^{S848A}-expressing cells is not due to a defect in exocytosis.

2. The conclusion that our findings reveal a new regulatory mechanism for Dyn2 activity in muscle cell that plays an important role in glucose metabolism and muscle physiology is totally unfounded as none of this is investigated in this study. Even if the GLUT4 traffic parameters were properly assessed, the above statement is an exaggeration when based solely on cell culture studies.

>> We agree with the reviewer and apologize for our overstatement. We have added new data to prove Dyn2S848 phosphorylation in mouse skeletal muscle, and further strengthened our conclusion by showing a beneficial effect of GSK3 α inhibitor in diet-induced insulin-resistant mice (Fig 8). In addition, we have also revised the sentence into “we identify a new role of GSK3 α in insulin-stimulated glucose disposal through regulating Dyn2-mediated GLUT4 endocytosis in muscle cells” (page 3 line 12-14).

3. The Introduction ends proposing a role of Dyn2 at the muscle T-tubules but the study is all performed in cell cultures that do not form T-tubules. Related to this, what is the relationship between the intracellular tubes shown in cells and T-tubules in a mature muscle?

>> We thank the reviewer for this important question. T-tubule in mature muscle cells result from the function of Bin1 on plasma membrane invagination which are evolutionary conserved from *Drosophila* to human. The intracellular tubules in our manuscript is derived from Bin1-GFP overexpression that mimics the T-tubule in a mature muscle cell (Lee et al., 2002, Science). We have introduced the role of Bin1 in T-tubule biogenesis in Introduction “Bin1 contains an N-BAR domain that forms a banana-shaped homodimer to sense, coordinate and generate membrane curvature, and is required for the biogenesis and maintenance of T-tubules in skeletal muscles” (page 6, line 3-5), and explained the rationale of the Bin1-GFP tubule we examined in this study: “The morphology of these membrane tubules generated by Bin1-GFP were reminiscent of T-tubules in terms of their high curvatures, and the enrichment of Bin1 and PI(4,5)P2 (Lee et al., 2002, Science). They could be severed into small vesicles by Dyn2, making them a feasible cellular system to validate Bin1-Dyn2 interactions observed in the reconstitution experiments (Chin et al., 2015, Laiman & Liu, 2020).” (page 11 line 6-11)

4. The subcellular fractionation and characterization suggest the P fraction has plasma membranes, since it expresses the Na/K-ATPase, but it also has a substantial amount of EEA1 contamination which is an early endosome marker. In addition, it is essential to show if any of the fractions contain TGN or recycling endosome markers or a GLUT4

vesicle marker such as VAMP2. More analysis and discussion of these fractionation results is warranted to support the conclusion that Dyn2-S848A is elevated at the PM relative to Dyn2-S848E.

>> We thank the reviewer for this helpful suggestion. By probing the TGN protein, Golgin-97, we obtained a more specific marker for light membrane indication (as shown in Fig 4A). This result together with the new HA-GLUT4-GFP data (Fig 4C-E) support the conclusion that Dyn2^{S848A} leads to higher ratio of GLUT4 on cell surface compared to Dyn2^{S848E} without affecting the total amount of GLUT4 in cells (Fig S3B-C).

5. Related to the above, the effect of Dyn mutants on total GLUT4 levels must be shown, as mislocalization of GLUT4 can result in its degradation at the lysosome.

>> Thanks for this important question, we hence have quantified the total GLUT4 levels in myotubes expressing different Dyn2 mutants as shown in Fig S3B and C, and found the expression of different Dyn2-S848 mutants does not significantly affect the total GLUT4 level in C2C12-derived myotubes.

6. The assignment of GLUT4-GFP puncta to the plasma membrane versus intracellular vesicles is not rigorously demonstrated. Typically, a fluorescent marker of the plasma membrane would be co-expressed with GLUT4-GFP or there are GLUT4 chimeric proteins with extracellular epitopes of HA or Myc that have been created for this purpose. This reviewer would insist that Figure 5B to G be repeated with a GLUT4-7myc-GFP construct that can be obtained from Addgene.

>> We agree with the reviewer and have utilized the recommended approach with HA-GLUT-GFP construct to repeat this experiment. The new data of fluorescent intensity ratio of extracellular HA-epitope/total GFP show similar result as previous approach we used. We thus have replaced the old result with the new one conducted by this better approach (Fig 4C-E).

7. Moreover, to be able to conclude that Dyn2 phosphorylation regulates GLUT4 endocytosis the authors should use the above construct to directly measure GLUT4 endocytosis, which would greatly solidify the conclusions of the effect of Dyn2WT and its mutant S848A and S848E.

>> We fully agree with the reviewer and have utilized the approach mentioned above (HA-GLUT4-GFP) to measure the effect of Dyn2 mutants on GLUT4 endocytosis. These new data are shown in Fig 4C-E.

8. What was the rationale for using C2C12 muscle cells for Transferrin uptake and L6 for GLUT4-GFP distribution? And why myotubes of the former but myoblasts of the latter?

>> We mainly use C2C12 cells for most of our experiments because of its better differentiation efficiency, as evident by the Bin1 induction after differentiation (Fig S3A). However, we switched cell lines from C2C12 to L6 myoblasts in order to study insulin-regulated GLUT4 trafficking, because C2C12 myocytes have been reported to lack an insulin-responsive vesicular compartment of GLUT4 (Tortorella and Pilch, 2002 *Am J Physiol. Endocrinol. Metab.*).

As for using cells at different stages, considering the detectable expression of Bin1 and its low transfection efficiency in L6 myoblasts, we did not further differentiate L6 into myotubes that normally takes about a week.

9. It would be essential to test if serum or insulin regulate Dyn2 phosphorylation in muscle cells. Related to this, what is the effect of inhibiting GSK3 α on transferrin uptake (or GLUT4 localization)?

>> We agree with the reviewer and have added a new result detecting Dyn2 phosphorylation in C2C12 myotubes that shows an increase of Dyn2S848 phosphorylation upon serum starvation and a reduction by insulin addition (Fig. 5D and E). For the effect of GSK3 α , we found that the kinase inactive GSK3 α ^{K148A} results in lower transferrin uptake and elevated surface ratio of HA-GLUT4-GFP (Fig 7A-D). We also observed similar effect in GLUT4 localization using GSK3 α inhibitor but not GSK3 β inhibitor in L6 myoblasts, reaffirming the isoform specific role of GSK3 α in endocytosis (Fig 7E-H).

10. Does the viral vector result in equivalent expression of GLUT4-GFP in all cells? Otherwise, it would be important to normalize the membrane and intracellular localization of GLUT4 to the expression in each cell. Related to this, myoblasts are rather thin cells, so that assigning surface to intracellular localization would highly benefit from showing confocal planes of analysis.

>> The concern about expression level of GLUT4-GFP is rightful, and we indeed have normalized the surface distributed GLUT4-GFP with the total GLUT4-GFP, and showed them as intensity ratio of GLUT4-GFP (surface/total) in previous version of this manuscript. To avoid this confusion in the new experiment, we have specified the Y-axis in Fig 4E into “surface HA-GLUT4-GFP (HA/GFP)”. In addition to the

quantification result, the representative confocal micrographs of surface and total HA-GLUT4-GFP (shown by anti-HA antibody staining without permeabilizing cells as well as the total GFP signal) are also shown in Fig 4C and D.

Minor issues

1. The authors state that C2C12 myotubes express more Bin1 than C2C12 myoblasts. It would be of interest to illustrate this with immunoblotting or qPCR. Along these lines, do L6 myoblasts express Bin1, could L6 myotubes express more Bin1, do L6 myoblasts express more Bin1 than C2C12 myoblasts on a per protein basis.

>> We thank the reviewer for this brilliant idea and have detected the expression of Bin1 in C2C12 and L6 at either myoblast or myotube stages with immunoblotting, as shown in Fig S3A. Indeed, L6 myoblasts express more Bin1 than C2C12 myoblasts. However, the differentiated L6 myotubes express less Bin1 than C2C12 myotubes, which is probably due to the low differentiation efficiency of L6 myoblast (Of note, we rarely obtained multi-nuclear myotubes from differentiated L6 myoblasts).

2. It is stated that the serum condition is physiological, but 5% serum is not precisely so.

>> We agree with the reviewer that cells cultured in medium containing 10% or 0% fetal bovine serum is indeed not physiological. We therefore have removed this description from the text.

3. Does the Dyn2 phospho-S848 antibody detect endogenous Dyn2 in L6? If so, some of the above experiments should be performed with the endogenous Dyn 2 (such as testing its phosphorylation status with serum or insulin).

>> We thank reviewer for this brilliant question! We therefore have tried many different cell types and stages together with Dyn2 phospho-S848 antibody optimization. We finally are able to detect the phosphorylation of endogenous Dyn2 in C2C12 myotubes (with proper amount of lysate and strict antibody blotting condition). Thus we are pleased and excited to show the effect of insulin signaling on endogenous Dyn2-Ser848 phosphorylation in C2C12 myotubes and mouse skeletal muscle Fig 5D-E, and 8E. Once again, thank you for this great suggestion!

4. Scansite 4.0 was used to discover GSK3alpha as a Dyn2-Ser848 kinase. It would be

of interest if the authors showed or described this analysis (including a citation of website hyperlink) in supplemental figures to demonstrate how highly ranked GSK3alpha appeared and if other kinases were also ranked highly.

>> We thank the reviewer for this helpful suggestion and have added the website hyperlink and the analysis result in Fig S4A and its legend.

Reviewer #2 (Comments to the Authors (Required)):

Laiman et al. present very interesting insights into dynamin functions and how these could be regulated in response to insulin signaling in the context of muscle cells. Using a combination of biochemistry and live cell imaging, the authors show that interactions between BIN1's SH3 domain and dynamin2's PRD inhibits dynamin2's capacity for membrane fission, explaining why BIN1-coated tubules in cells are resilient to severing by dynamin. Furthermore, the authors address reasons why certain CNM-specific mutants of BIN1 appear to be organized not as tubules but as vesicles in cells. These CNM-specific mutants, which result in a partial truncation of its SH3 domain, appear to weaken interactions with dynamin2 and consequently get severed by dynamin. The authors then suggest a physiological signaling axis managed by insulin dependent GSK3alpha that phosphorylates dynamin at its PRD on S848, which weakens its ability to interact with BIN1 and facilitates scission of BIN1 tubules. The authors extend these results to understand how such a kinase-controlled partner interaction influences dynamin's functions during CME in muscle cells using GLUT4 as a marker cargo. The same insulin-dependent signaling apparently contributes to facilitating GLUT4 endocytosis via CME.

The results described in the manuscript can be divided into two parts, which are not necessarily linked. The first addresses BIN1-dynamin interactions while the second address dynamin's role in CME. The experiments carried out for the first part are well-designed, meticulous and convey novel information on dynamin's function with respect to the stability of BIN1 tubules. The data presented on the mechanism for why the CNM-specific mutants of BIN1 are quite significant and could have a profound impact on understanding the pathology of CNM. Regrettably, I find the second part lacking a clear rationale and could benefit from more pointed experiments and rigorous analyses.

>> We thank the reviewer for pointing out the weakness in our previous manuscript. We agree with the reviewer and, given the novel effect of GSK3 α on endocytosis, have carefully re-framed our manuscript to focus on the regulation of GLUT4 endocytosis

in muscle. Therefore, we simplified the biochemical results of Bin1-Dyn2 interaction, and added several new data to support the role of GSK3 α in regulating GLUT4 endocytosis in muscle cells (Fig 7C-H). Additionally, we applied our finding in animal experiments to show that pharmacological inhibition of GSK3 α in diet-induced insulin-resistant mice leads to better insulin and glucose response (Fig 8). With these new data and the novelty of isoform specific function of GSK3 α on endocytosis, we have revised the title of this manuscript into “GSK3 α , but not GSK3 β , phosphorylates dynamin-2 to promote GLUT4 endocytosis in muscle cells”.

While this work presents novel insights into why certain CNM-associated BIN1 mutants display an altered vesicular morphology, I was unsure of the exact mechanism for these effects. The mutations cause a partial truncation of the SH3 domain in BIN1 but surprisingly, they seem to fare as well as an SH3 deletion construct of BIN1 in fission assays. How do the authors explain this? The authors state that a relatively small decline in binding affinity between SH3-PRD interactions is sufficient to rescue dynamin function but it would be useful to estimate the binding affinities. In other words, how much of a lowering in binding affinity would be necessary to strike a balance between efficient recruitment of dynamin to BIN1-coated tubules without causing an inhibition in its fission activity?

>> Thanks for this intriguing question. Although the SH3 domain truncation and deletion of Bin1 both promote Dyn2 fission activity in vitro, the SH3 deletion mutant has more pronounced effect than the Bin1^{Q434X} and ^{K436X} (Fig 1B and 2F). Therefore, the partially truncated BIN1 does not fare as well as a BIN1 Δ SH3 in fission assays.

To measure the binding affinity between Bin1^{K436X}-Dyn2 compared with Bin1^{WT}-Dyn2, we used GST pulldown assay. We found the dissociation constant K_d of Bin1^{K436X} (the mutant with 19 aa truncation) compared to Bin1^{WT} toward Dyn2 is slightly increased to 267 ± 70 nM from 180 ± 60 nM (response figure I, below), yet the proteoliposomes from Bin1^{K436X} co-assembly with Dyn2 were significantly smaller in diameter (Fig 2G and H). Although we did not include this data in manuscript in order to focus on GLUT4 endocytosis, the impact of binding affinity between Dyn2 and SH3 proteins is of great interest. We believe it is worthy to solve the structure of Bin1-Dyn2^{S848E} assembled on liposome.

Response figure i. Binding affinity between Bin1 and the PRD of Dyn2. Increasing concentrations of GST-PRD was used to pull down 0.2 μM purified Bin1^{WT} or Bin1^{K436X}. PRD-bound Bin1 was then analyzed.

The dissociation constant (K_d) was determined by curve fitting using nonlinear regression and shown with SD ($n = 3$).

The suggested model that insulin-dependent signaling regulates dynamin functions is indeed interesting. In the context of BIN1, this link is largely inferred from the use of phosphomimetic mutants of dynamin2, which is understandable for biochemical experiments. But establishing that such signaling indeed causes a partial disengagement of dynamin from BIN1 tubules in cells and a subsequent severing of the tubule could provide strong validation of this model. This can be easily assayed by monitoring the effect of an acute application of insulin to cells followed by the two-color imaging of BIN1 and dynamin. Was this checked?

>> We thank the reviewer for this exciting suggestion. Accordingly, we have conducted a time-lapse imaging analysis of cells co-transfected with Dyn2-mCherry and Bin1-GFP in the presence or absence of insulin. In line with our hypothesis, the Bin1-GFP-coated tubules displayed significant fragmentation and vesiculation in the absence of insulin, which could be rescued by 1 hour-treatment of insulin to cells (Fig 5F-G).

The in vitro data on fission manifests from dynamin's ability to vesiculate the planar SUPER templates. Quite likely, such vesiculation relies on an upstream process that causes tubules to grow from the template which are then captured and severed by dynamin. Dynamin1's strong tendency to self-assembly causes both tubulation and severing of the membrane. But dynamin2 requires other tubulating proteins such as endophilin and BIN1. How well does this assay recapitulate cellular physiology? Is the growth of BIN1 tubules in cells influenced by the presence of endogenous dynamin? Clearly BIN1 inhibits dynamin-catalyzed vesiculation from SUPER templates but does a complex of BIN1 and dynamin even tubulate SUPER templates? Some of these points could be clarified in the manuscript.

>> Thanks for these inspiring questions.

Firstly, how well does this assay recapitulate cellular physiology?

The supported bilayer with excess membrane reservoir (SUPER) template is a powerful tool to study membrane tubulation and vesiculation ability of purified proteins *in vitro* (Pucadyil and Schmid, 2008, Cell). However, this assay could not completely recapitulate the fission requirement *in vivo*. For example, dynamin-1 can catalyze membrane fission from SUPER template without PRD, but it could not complete this function in cells without PRD, the domain important for dynamin recruitment onto plasma membrane. Therefore, this assay could recapitulate the regulation of dynamin fission activity to a reasonable degree, but the experimental results need to be validated and confirmed carefully in cells.

>> Secondly, is the growth of BIN1 tubules in cells influenced by the presence of endogenous dynamin?

Dynamin has been reported to inhibit the membrane tubulation activity of BAR-domain containing proteins *in vitro* (Neumann et al. 2013 JCB). Indeed, when purified Bin1 and Dyn2 proteins were co-incubated with SUPER template, Dyn2 displayed clear dose-dependent inhibitory effect on the tubulation ability of Bin1 (response figure II, below). We also find that this inhibitory effect depends on the binding affinity between Dyn2 and Bin1 in which Dyn2^{S848E} shows less inhibitory ability on Bin1 tubulation. To examine the relative expression level of endogenous Dyn2 and Bin1, we have quantified their protein level in C2C12 myotubes by comparing the Westernblot band intensity with purified proteins, and found that Bin1 is five times more abundant (molar ratio) than Dyn2 in C2C12 myotubes.

Together, the Dyn2-Bin1 complex at 1:1 ratio could not tubulate SUPER template efficiently; yet the endogenous Dyn2 would only have limited effect on the tubulation ability of Bin1 in myotubes due to the high expression level of Bin1.

Response figure ii. Membrane tubulation of Bin1 in the presence of different Dyn2 mutants. 0.5 μ M wild-type Bin1 was incubated with SUPER templates together with different amounts or mutants of Dyn2 for 10 min at room temperature. The tubulation of SUPER templates was observed and imaged under fluorescent microscopy. Scale, 5 μ m.

The part about insulin-dependent regulation of dynamin function in GLUT4 trafficking appears quite preliminary and the suggested model is tenuous at best. Based on how the manuscript is written, I'm not sure if the authors are trying to emphasize a common link between dynamin's function in BIN1 tubule dynamics and in CME. If yes, then isoform 8 of BIN1 does not have any clathrin and AP2 binding sites so it's unlikely to even be recruited to CCPs. Such a link therefore lacks justification at the level of participant proteins. The effects of insulin signaling on CME should therefore manifest from a regulation of dynamin's interaction with some other binding partner. But as the authors themselves convince us so nicely of the fact that not all SH3 interactions are inhibitory to dynamin's function in fission, even a causal link appears vague at best.

>> We thank the reviewer for this comment and have substantially improved the section regarding insulin-dependent regulation of Dyn2 in mediating GLUT4 endocytosis. We quantified the surface/total GLUT4 ratio, both the endogenous and exogenously expressed, in C2C12 and L6 myoblasts (Fig 4). Moreover, we have demonstrated that the kinase inactive $GSK3\alpha^{K148A}$ results in elevated surface ratio of HA-GLUT4-GFP (Fig 7C, D). We also observed similar effect in GLUT4 localization using GSK3 α inhibitor but not GSK3 β inhibitor, reaffirming the isoform specific role of GSK3 α in muscle cells (Fig 7E-H).

Secondly, regarding the point that Bin1 isoform 8 does not contain clathrin and AP-2 binding (CLAP) domain, Bin1 probably affect the function of Dyn2 on CME through protein binding and sequestration. The negative regulatory role of Bin1 on Dyn2-mediated endocytosis fits with previous reports of increased transferrin uptake in cells with downregulation of ubiquitous Bin1 lacking the CLAP domain (Muller et al., 2003, Mol Cell Biol and Pant et al., 2009, Nat Cell Biol). Distinct from transferrin that is internalized into the cell mainly through CME, GLUT4 is endocytosed into muscle cells via CME and clathrin- and caveolae-independent (CCI) pathway that probably occurs at the sarcolemma and the T-tubule, respectively (Antonescu et al., 2008, Traffic; Marette et al., 1992, JCB).

In sum, I think the manuscript has some very nice data, especially those that pertain to the CNM-specific mutants of BIN1. I believe these would generate quite a bit of excitement in the community. I would recommend a shorter but more focused manuscript, emphasizing the BIN1-dynamin interactions addressing the experiments and clarifications to text suggested above.

>> We appreciate the reviewer for the recognition on the biochemical analysis on Bin1 mutants. However, because the negative role of Bin1 on Dyn2 has been reported in mouse (Cowling et al., 2017, JCI; Lionello et al., 2022, PNAS), we think the main

novelty of our work is the Dyn2-S848 phosphorylation by GSK3 α and its impact on endocytosis regulated by insulin signaling. Therefore, we worry that a shorter manuscript focusing on CNM-specific mutant Bin1 could not provide enough scientific merit for Journal of Cell Biology. To avoid confusion, we have carefully addressed the questions the reviewer mentioned and prepared a better-supported and well-explained manuscript focusing on how GSK3 α , but not GSK3 β , regulates GLUT4 endocytosis in muscle. We hope the reviewer will find the new manuscript well-supported and convincing.

Reviewer #3 (Comments to the Authors (Required)):

Laiman and colleagues propose experiments and data showing the interplay between Bin1, Dyn2 and GSK3 in vitro and in cells, and the impact on membrane tubulation and on GLUT4 endocytosis. As Bin1 and Dnm2 are mutated in a human disease, they also convincingly decipher the impact of some of the mutations. They discovered a regulation of Dyn2 by phosphorylation through GSK3, and a 'bi-directional' regulation by Bin1 where the N-BAR domain of Bin1 favors Dyn2 membrane fission while the SH3 domain inhibits Dyn2-mediated fission.

Several experiments revealed such mechanism appears specific to Bin1, at least not conserved with endophilin, which is very interesting. Similarly the experimental and literature data comparing the regulation of Dyn1 and Dyn2, especially through different GSK3 isoforms, are exciting and should most probably be of interest for future studies. The concept of Bin1 negatively regulating Dyn2 was shown before in vitro and in vivo including in mammals, as the effect of the SH3 mutations on Dyn2 binding, and is thus not brand new. However, as the authors pointed, they made a great job to decipher the underlying molecular mechanism concerning this part. The regulation of GLUT4 trafficking through GSK3-mediated Dyn2 848 phosphorylation appears entirely novel. >> We are very grateful for the reviewer's helpful suggestions and enthusiasm to our work. More specific responses to the reviewer are listed below.

Major points:

-phosphorylation status of Dyn2 WT in several experiments:

fig4D: it is interesting but unclear to me how/why S848A leads to a Bin1-mediated inhibition of Dyn2 fission activity. In Fig4C there seems to be an increased Bin1 binding (compared to Dyn2 WT) that would fit but how this is happening? Probably

the authors have envisaged Dyn2 produced from sf9 cells is partially phosphorylated : they can test that with their antibody. Also in the other experiments, did they use phosphatases or inhibitors in their Dyn2 preparation to be sure they were performed with a Dyn2-WT with known phosphorylation status?

Similarly, in transfected cells, is the transfected Dyn2 WT partially phosphorylated; that sounds feasible as the S848A fissions less.

Similarly in Fig5 GLUT4 ratio, Dyn2 WT seems to have intermediate values between the non-phosphorylated 848A and the pseudo-phosphorylated 848E, again suggesting a partial phosphorylation that could have been demonstrated or rule out with inhibitors and their anti-phospho antibodies (either p-S848-Dyn2 or as in Fig6A).

>> We agree with the reviewer that Dyn2^{WT} should be partially phosphorylated in cells when GSK3 α is active. Indeed, the transfected HA-Dyn2^{WT} or endogenous Dyn2 is partially phosphorylated which could be enhanced upon serum starvation in L6 myoblasts or C2C12 myotubes, or in the presence of constitutive active GSK3 α ^{S21A} in HeLa cells (Fig 5A-E, and 6A-B).

However, because we did not use phosphatase inhibitors during protein purification, we think it is unlikely that the wild-type Dyn2 purified from Sf9 cells remained partially phosphorylated. In line with that, the purified Dyn2^{WT} protein did not display significant p-S848 signal in Fig 6E without GSK3 α addition. We therefore reason the lower fission activity of Dyn2^{S848A} than Dyn2^{WT} in the presence of Bin1 in vitro is probably due to the slightly higher binding affinity between Dyn2^{S848A} with Bin1 (Fig 3B and C). The alanine is a hydrophobic amino acid, whereas serine and glutamate is a polar and charged amino acid, respectively. We thus speculate the hydrophobic alanine may result in a slight increase of Bin1 binding with Dyn2^{S848A} and higher inhibitory effect of Bin1 on the fission activity of Dyn2^{S848A} than on the unphosphorylated, purified Dyn2^{WT} (Fig 3D).

-Concerning the experiments testing the role of the PI domain and of mutations truncating the SH3, how do they impact on Bin1 conformation and does this impact on the regulation of Dyn2 ? Indeed Kojima et al 2004 suggested the PI domain binds the SH3 domain of Bin1. How do the authors take into account or test that the deletion of the PI or SH3 domains unmasks one of these domains ? This Bin1 autoinhibition appears an important mechanism that might alter the conclusion of the experiments investigating the regulation of Dyn2 functions.

On a similar line, they found the deletion of the PI domain, supposedly 'opening' Bin1, has no effect on Dyn2 fission activity. Can they study this point with pulldown as in

figs 2 and 4 (is PI domain altering Dyn2 binding through itself or the SH3).

>> We thank the reviewer for these intriguing questions. Indeed, the intramolecular interaction between PI and SH3 domains of Bin1 has been demonstrated to result in Bin1 autoinhibition. Therefore, the direct interaction between dynamin and Bin1 could only be observed in the presence of liposome, indicating the binding of PI motif to lipids could free the SH3 domain for PRD binding (Kojima et al., 2004, EMBO J; Yoshida et al., 2004, EMBO J). On the other hand, the presence of Dyn2 also enhances the binding of Bin1 to liposome (Fig 1F). Thus, it is necessary to add liposome into the reaction for analyzing interaction between full length Bin1 and Dyn2 (Fig 2B-C).

To analyze the effect of PI motif on Dyn2 binding with Bin1, we have conducted GST-PRD pull down assay to test the binding between Dyn2-PRD and Bin1 with or without PI domain in the **absence** of liposome. In line with Bin1 autoinhibition resulting from PI-SH3 interaction, the full length Bin1 could not bind to Dyn2 without liposome, and the deletion of PI motif significantly improves it (response figure III, below). On the other hand, the presence of SUPER template relieves Bin1 autoinhibition, thus resulting in indistinguishable effect between Bin1 and Bin1 Δ PI on Dyn2 fission activity (Fig S1E).

Response figure iii. Effect of PI motif on Bin1-Dyn2 interaction. Different GST-tagged Bin1 proteins (0.5 μ M) were incubated with 0.3 μ M Dyn2, then the bound Dyn2 was detected and quantified by SDS-PAGE electrophoresis and CBR staining. The data was quantified and shown (n = 3).

In order to focus on the isoform-specific role of GSK3 α on endocytosis, we did not include this data into this manuscript though the effect of PI motif on Bin1-Dyn2 interaction is very interesting.

Minor points:

-is there a way to quantify results of figure 1F, as the helix organization seems to be rather preserved with Bin1+ Dyn2, to definitely conclude Bin1 alters the assembly of membrane-bound Dyn2 (in term of spiral pattern, not on tubules diameter that is reported in figure 1G).

>> We thank the reviewer for this constructive suggestion. We have measured the distance between Dyn2 helix (pitch) with or without Bin1, and find the pitch of Dyn2 helix to be significantly wider in the presence of Bin1. This result is shown in Fig S1C.

-is there a statistical increase in Dyn2 fission with N-BAR-PI compared to Bin1 in fig S2B as in Fig 1 ? or none of the conditions are statistically different ?

>> Thanks for the suggestion. We have added the statistical analysis result into the figure. The fission activity of Dyn2 is significantly higher in the presence of N-BAR-PI than the Bin1^{WT} (Fig S1E).

-FigS2H: it will be better to compare +/-PI domain with the same tag as the use of different tags may impact on potential dimerization; to rule out BIN1-GFP tubulates due to GFP and not because of the addition of the PI domain.

>> We totally agree with the reviewer that it is better to test the effect of PI domain with the same tag and apologize for not being meticulous. We thus have generated a Bin1-ΔPI-GFP and repeat this experiment as shown in Fig S1F.

-FigS3C: tubules look thicker with the mutants in cells; any confirmation of that, or was it similar on liposomes for Bin1 alone ?

>> We believe the CNM-Bin1 mutant tubules in cells are in similar diameter, and they looked thicker in some regions because several tubules are aligned together. To avoid confusion, we have chosen other representative images with less tubule clustering (Fig 2I).

-Fig3C : in the time lapse, if possible it will be more informative to show Dyn2-mCherry together with Bin1-GFP, to assess if fission is happening upon recruitment of Dyn2.

>> We thank the reviewer for this helpful suggestion and have added Dyn2-mCherry channel into the new Fig 2L. In addition, though the fission of Bin1-GFP tubule is happening in the presence of Dyn2-mCherry, the fission does not happen immediately after the recruitment of Dyn2-mCherry onto Bin1-GFP tubule (response figure IV). This observation suggests that fission of Bin1 tubule is driven by the rearrangement of Dyn2-Bin1 complex probably after the phosphorylation of Dyn2-S848.

response figure iv. Spatiotemporal distribution of Dyn2-mCherry on a fission event at Bin1-GFP tubule. Time-lapse representative images of Bin1-GFP co-expressed with Dyn2-mCherry in C2C12 myoblast were magnified and shown. White arrow heads indicate the occurrence of membrane fission. Scale, 2 μ m.

-Fig4 and associated data: the rationale to select the 848 and 856 sites may be clarified as the present findings seem to be based on the description of the 848 phosphorylation in Efendiev 2002. However there are other PXXP motif nearby that were shown to bind to amphiphysin but were not tested (eg QIPSRPVR). Do the authors think or have shown that phosphorylation at these other sites is not implicated in Bin1 binding and regulation.

>> Thanks again for this insightful question. Currently, we do not know whether the QIPSRPVR motif that Amphiphysin binds with (Solomaha et al., 2005, JBC) is phosphorylated due to the difficulty in detecting phosphorylation of endogenous or exogenously expressed Dyn2 by Mass Spectrometry. We have tried several different approaches yet have not nailed it. We think it is very likely that phosphorylation at other sites is also involved in Bin1 binding and regulation of Dyn2. Thus it is worthy to comprehensively investigate the phosphorylation sites and kinases of Dyn2.

-Fig5C-D: the way to quantify surface GLUT4 with the intensity ratio surface/total is assuming only dot-like pattern of GLUT4-GFP is intracellular; subcellular fractionation as in 5A might probably be more precise.

>> We agree with the reviewer. Accordingly, we have utilized GLUT4 chimeric proteins equipped with extracellular epitopes of HA to better define the effect of Dyn2 phosphorylation and GLUT4 endocytosis (Fig 4C-E). We hope the reviewer will find the new method with confocal microscopy data clear and convincing.

-top of p19: may precise 'without affecting the extent of phosphorylation of Dyn2-848 MUTANT' (or 848A)

>> We thank the reviewer for the correction and have corrected it into “without affecting the extent of phosphorylation of HA-Dyn2^{S848A}” (page 16, line 1-2).

-Discussion: The authors conclude that disruption of t-tubule homeostasis is the common hallmark of CNM. They also showed impact on CME and GLUT4; could they speculate on the impact of these defects in the disease.

>> We thank the reviewer for the helpful suggestion. Despite the contradictory results reported for the effect of Dyn2-CNM mutants expression on CME in different cell lines (Bitoun et al., 2009, Hum Mutat; Durieux et al., 2010, Hum Mol Genet; Koutsopoulos et al., 2011, PLoS One; Liu et al., 2011, Traffic), abnormal perinuclear accumulation of GLUT4 has been observed in muscle biopsies from CNM patients with R465W and R369Q Dyn2 mutations (González-Jamett et al., 2017, Scientific Reports). Therefore, even though currently no abnormality in blood glucose from the clinical data of CNM patients has been reported to our best knowledge, we speculate that CME and the surface level of GLUT4 could also be affected in the skeletal muscle of patients. We have included this speculation in Discussion (page 19, line 17-18).

-Please clarify in methods which Dyn2 splice isoform was used as the Schmid's team and others showed they have different functions

>> We have clarified in the methods that the Dyn2 used in this study is isoform 1 (page 22, line 6).

-In general it would help to indicate the number of biological replicates in each legends.

>> We have indicated the number of biological replicates in each legends.

October 12, 2022

RE: JCB Manuscript #202102119R-A

Prof. Ya-Wen Liu
National Taiwan University
Institute of Molecular Medicine
No. 1, Sec. 1, Jen-Ai Rd., R1517
Taipei 10002
Taiwan

Dear Prof. Liu:

Thank you for submitting your revised manuscript entitled "GSK3 α , but not GSK3 β , phosphorylates dynamin-2 to promote GLUT4 endocytosis in muscle cells". Two of the original reviewers have now assessed your revised manuscript and, as you can see, they are satisfied with revisions. We would be happy to publish your paper in JCB pending final revisions necessary to meet our formatting guidelines (see details below). In your final revision, please be sure to address the reviewers' remaining concerns with appropriate text edits.

To avoid unnecessary delays in the acceptance and publication of your paper, please read the following information carefully. Please go through all the formatting points paying special attention to those marked with asterisks.

A. MANUSCRIPT ORGANIZATION AND FORMATTING:

Full guidelines are available on our Instructions for Authors page, <https://jcb.rupress.org/submission-guidelines#revised>.
Submission of a paper that does not conform to JCB guidelines will delay the acceptance of your manuscript.

1) Text limits: Character count for Articles and Tools is < 40,000, not including spaces. Count includes title page, abstract, introduction, results, discussion, and acknowledgments. Count does not include materials and methods, figure legends, references, tables, or supplemental legends.

2) Figures limits: Articles and Tools may have up to 10 main text figures.

*** Please note that main text figures should be provided as individual, editable files.

3) Figure formatting:

Molecular weight or nucleic acid size markers must be included on all gel electrophoresis.

*** Scale bars must be present on all microscopy images, including inset magnifications. Please include scale bars in main Figs 1H (inset magnifications), 2E (inset magnifications), 2G (inset magnifications), 2I-J (inset magnifications), 2L (right column micrographs), 3E (inset magnifications), 4C-D (right column micrographs), 5F (inset magnifications), 7C (bottom micrographs), 7E (bottom micrographs), 7G (bottom micrographs) and supplemental Figs. 1E (inset magnifications), 2D (inset magnifications).

*** Also, please avoid pairing red and green for images and graphs to ensure legibility for color-blind readers. If red and green are paired for images, please ensure that the particular red and green hues used in micrographs are distinctive with any of the colorblind types. If not, please modify colors accordingly or provide separate images of the individual channels.

4) Statistical analysis:

Error bars on graphic representations of numerical data must be clearly described in the figure legend.

*** The number of independent data points (n) represented in a graph must be indicated in the legend. Please, indicate 'n' for panel 2H in the corresponding legend.

*** Please, indicate whether 'n' refers to technical or biological replicates (i.e. number of analyzed cells, samples or animals, number of independent experiments).

If independent experiments with multiple biological replicates have been performed, we recommend using distribution-reproducibility SuperPlots (please, see Lord et al., JCB 2020) to better display the distribution of the entire dataset, and report statistics (such as means, error bars, and P values) that address the reproducibility of the findings.

Statistical methods should be explained in full in the materials and methods in a separate section.

For figures presenting pooled data the statistical measure should be defined in the figure legends.

Please also be sure to indicate the statistical tests used in each of your experiments (both in the figure legend itself and in a separate methods section) as well as the parameters of the test (for example, if you ran a t-test, please indicate if it was one- or two-sided, etc.).

*** As you used parametric tests in your study (i.e. t-tests), you should have first determined whether the data was normally distributed before selecting that test. In the stats section of the methods, please indicate how you tested for normality. If you did not test for normality, you must state something to the effect that "Data distribution was assumed to be normal but this was not formally tested."

5) Abstract and title:

The abstract should be no longer than 160 words and should communicate the significance of the paper for a general audience.

*** The title should be less than 100 characters including spaces. Make the title concise but accessible to a general readership. To convey the advance more clearly, we suggest the following title: "GSK3 α phosphorylates dynamin-2 to promote GLUT4 endocytosis in muscle cells"

6) Materials and methods:

*** Should be comprehensive and not simply reference a previous publication for details on how an experiment was performed. Please provide full descriptions in the text (at least in brief) for the "Protein expression and purification", "Preparation of lipid templates", "In vitro fission assays", "Subcellular fractionation and analysis of surface:total GLUT4 ratio" sections for readers who may not have access to referenced manuscripts. The text should not refer to methods "...as previously described."

Also, the materials and methods should be included with the main manuscript text and not in the supplementary materials.

7) Please be sure to provide the sequences for all your primers/oligos and RNAi constructs in the materials and methods.

You must also indicate in the methods the source, species, and catalog numbers (where appropriate) for all your antibodies.

8) Microscope image acquisition:

The following information must be provided about the acquisition and processing of images:

a. Make and model of microscope

b. Type, magnification, and numerical aperture of the objective lenses

c. Temperature

*** d. imaging medium

e. Fluorochromes

f. Camera make and model

g. Acquisition software

h. Any software used for image processing subsequent to data acquisition. Please include details and types of operations involved (e.g., type of deconvolution, 3D reconstitutions, surface or volume rendering, gamma adjustments, etc.).

10) Supplemental materials:

There are strict limits on the allowable amount of supplemental data. Articles/Tools may have up to 5 supplemental figures. There is no limit for supplemental tables.

*** Please note that supplemental figures and tables should be provided as individual, editable files.

*** A summary of all supplemental material should appear at the end of the Materials and Methods section (please see any

recent JCB paper for an example of this summary).

11) eTOC summary:

*** A ~40-50 word summary that describes the context and significance of the findings for a general readership should be included on the title page. The statement should be written in the present tense and refer to the work in the third person. It should begin with "First author name(s) et al..." to match our preferred style.

12) Conflict of interest statement:

JCB requires inclusion of a statement in the acknowledgements regarding competing financial interests. If no competing financial interests exist, please include the following statement: "The authors declare no competing financial interests."

13) A separate author contribution section is required following the Acknowledgments in all research manuscripts.

*** All authors should be mentioned and designated by their first and middle initials and full surnames and the CRediT nomenclature is encouraged (<https://casrai.org/credit/>).

14) ORCID IDs: ORCID IDs are unique identifiers allowing researchers to create a record of their various scholarly contributions in a single place. At resubmission of your final files, please consider providing an ORCID ID for as many contributing authors as possible.

15) Materials and data sharing:

All animal and human studies must be conducted in compliance with relevant local guidelines, such as the US Department of Health and Human Services Guide for the Care and Use of Laboratory Animals or MRC guidelines, and must be approved by the authors' Institutional Review Board(s). A statement to this effect with the name of the approving IRB(s) must be included in the Materials and Methods section.

*** As a condition of publication, authors must make protocols and unique materials (including, but not limited to, cloned DNAs; antibodies; bacterial, animal, or plant cells; and viruses) described in our published articles freely available upon request by researchers, who may use them in their own laboratory only. All materials must be made available on request and without undue delay. Please, indicate whether the cell lines, plasmids and reagents generated in this study have been deposited in public repositories. If not, please state that they would be made available to the scientific community upon request in the 'Data availability' section.

All datasets included in the manuscript must be available from the date of online publication, and the source code for all custom computational methods, apart from commercial software programs, must be made available either in a publicly available database or as supplemental materials hosted on the journal website. Numerous resources exist for data storage and sharing (see Data Deposition: <https://rupress.org/jcb/pages/data-deposition>), and you should choose the most appropriate venue based on your data type and/or community standard. If no appropriate specific database exists, please deposit your data to an appropriate publicly available database.

16) Please note that JCB now requires authors to submit Source Data used to generate figures containing gels and Western blots with all revised manuscripts. This Source Data consists of fully uncropped and unprocessed images for each gel/blot displayed in the main and supplemental figures. The Source Data files will be directly linked to specific figures in the published article.

Since your paper includes cropped gel and/or blot images, please be sure to provide one Source Data file for each figure that contains gels and/or blots along with your revised manuscript files. File names for Source Data figures should be alphanumeric without any spaces or special characters (i.e., SourceDataF#, where F# refers to the associated main figure number or SourceDataFS# for those associated with Supplementary figures). The lanes of the gels/blots should be labeled as they are in the associated figure, the place where cropping was applied should be marked (with a box), and molecular weight/size standards should be labeled wherever possible.

B. FINAL FILES:

Thank you for this interesting contribution, we look forward to publishing your paper in Journal of Cell Biology.

Sincerely,

Satyajit Mayor
Monitoring Editor
Journal of Cell Biology

Lucia Morgado-Palacin, PhD
Scientific Editor
Journal of Cell Biology

Reviewer #1 (Comments to the Authors (Required)):

The authors are commended for having completely revamped the manuscript in attending to the reviewers' comments. In particular, they have used HA-GLUT4-GFP in a number of assays and the data is good and reliable. The imaging is of high quality. The added information presented in the new Figures 4, 6 and 8, and in parts of Figure 5 attend to the questions asked in the review in a very convincing way. Fig 9 is a good model figure that will convey the model in a clear way. The use of a GSK3alpha inhibitor to improve GTT and ITT in HFat/HSuc fed mice is intriguing and speaks to the physiological significance of the study, which is now very convincing.

A few brief issues could be commented upon to finalize the review:

1. AMPK is thought to possibly slow GLUT4 endocytosis in cardiomyocytes and skeletal muscle, but the evidence is not strong or has not delved into mechanism. Could AMPK phosphorylate S848 on Dyn2 or could there be an unappreciated role for GSK3alpha in exercise or contraction -stimulated glucose uptake in muscle tissue? Could the authors speculate in the Discussion?

2. Beg, Katome and Leto are cited in several places as studies that show Akt is required for GLUT4 translocation, but one is a review and the others focus on this pathway in adipocytes. Please quote the literature pertaining to the relationship of Akt and GLUT4 translocation, in response to insulin, in muscle tissue and L6 cells.
3. There are typos in the bar colour key of the Figure 1B. There are a couple of mM that need to be changed to microM (μ M).

Reviewer #2 (Comments to the Authors (Required)):

Just as authors are expected to provide a point-wise response to the reviewers' comments, I feel incumbent to do the same and provide below a point-wise response to the authors' response.

I has indicated earlier that the results described in the manuscript can be divided into two parts, which are not necessarily linked. The first addresses BIN1-dynamin interactions while the second address dynamin's role in CME. The experiments carried out for the first part are well-designed, meticulous and convey novel information on dynamin function with respect to the stability of BIN1 tubules. The data presented on the mechanism for why the CNM-specific mutants of BIN1 are quite significant and could have a profound impact on understanding the pathology of CNM. Regrettably, I find the second part lacking a clear rationale and could benefit from more pointed experiments and rigorous analyses.

In response to this comment (and from those of the other reviewers), the authors have done a superb job of modifying their manuscript. The revised manuscript is more focused and explicit in informing the community of a regulatory mechanism by which insulin-dependent signalling via GSK3 α facilitates endocytosis of GLUT4 via relief of inhibition of dyn2 by BIN1.

I had indicated earlier that while this work presents novel insights into why certain CNM-associated BIN1 mutants display an altered vesicular morphology, I was unsure of the exact mechanism for these effects. The mutations cause a partial truncation of the SH3 domain in BIN1 but surprisingly, they seem to fare as well as an SH3 deletion construct of BIN1 in fission assays. How do the authors explain this? The authors state that a relatively small decline in binding affinity between SH3-PRD interactions is sufficient to rescue dynamin function but it would be useful to estimate the binding affinities. In other words, how much of a lowering in binding affinity would be necessary to strike a balance between efficient recruitment of dynamin to BIN1-coated tubules without causing an inhibition in its fission activity?

In response to this comment, the authors have carried out additional experiments and measured binding affinities between BIN1 and dyn2 PRD. I'm not sure if the apparent 1.5-fold reduction in binding affinity of the GST PRD of dyn2 with BIN1(WT) and BIN1(K346x) is significant. The regression fits certainly don't appear so. This could indicate that it's not just a reduction in binding that renders BIN1(K346x) to display its phenotype in cells. But in the context of the revised manuscript, I would not hold it against the authors on this point. This could perhaps be considered as an additional aspect for the authors to work upon in a different manuscript.

I had indicated earlier that the in vitro data on fission manifests from dynamin's ability to vesiculate the planar SUPER templates. Quite likely, such vesiculation relies on an upstream process that causes tubules to grow from the template which are then captured and severed by dynamin. Dynamin1's strong tendency to self-assembly causes both tubulation and severing of the membrane. But dynamin2 requires other tubulating proteins such as endophilin and BIN1. How well does this assay recapitulate cellular physiology? Is the growth of BIN1 tubules in cells influenced by the presence of endogenous dynamin? Clearly BIN1 inhibits dynamin-catalyzed vesiculation from SUPER templates but does a complex of BIN1 and dynamin even tubulate SUPER templates? Some of these points could be clarified in the manuscript.

In response to this comment, the authors state that "supported bilayer with excess membrane reservoir (SUPER) template is a powerful tool to study membrane tubulation and vesiculation ability of purified proteins in vitro (Pucadyil and Schmid, 2008, Cell). However, this assay could not completely recapitulate the fission requirement in vivo. For example, dynamin-1 can catalyze membrane fission from SUPER template without PRD, but it could not complete this function in cells without PRD, the domain important for dynamin recruitment onto plasma membrane. Therefore, this assay could recapitulate the regulation of dynamin fission activity to a reasonable degree, but the experimental results need to be validated and confirmed carefully in cells. Secondly, is the growth of BIN1 tubules in cells influenced by the presence of endogenous dynamin? Dynamin has been reported to inhibit the membrane tubulation activity of BAR-domain containing proteins in vitro (Neumann et al. 2013 JCB). Indeed, when purified Bin1 and Dyn2 proteins were co-incubated with SUPER template, Dyn2 displayed clear dose-dependent inhibitory effect on the tubulation ability of Bin1 (response figure II, below). We also find that this inhibitory effect depends on the binding affinity between Dyn2 and Bin1 in which Dyn2S848E shows less inhibitory ability on Bin1 tubulation. To examine the relative expression level of endogenous Dyn2 and Bin1, we have quantified their protein level in C2C12 myotubes by comparing the Western blot band intensity with purified proteins and found that Bin1 is five times more abundant (molar ratio) than Dyn2 in C2C12 myotubes. Together, the Dyn2-Bin1 complex at 1:1 ratio could not tubulate SUPER template efficiently; yet the endogenous Dyn2 would only have limited effect on the tubulation ability of Bin1 in myotubes due to the high expression level of Bin1."

I think this again is an important point which the authors could consider for a follow-up manuscript. That the relative abundance

of dyn2 and BIN1 is important for this signalling axis to function is an important point. This is rarely considered by the community when phenotypic results are analyzed by overexpressing BIN1.

Minor comments:

In the abstract, perhaps the authors could consider changing 'unleashing' to 'relieving' in the sentence 'In the absence of insulin, GSK3 α phosphorylates Dyn2 to unleash the inhibition of Bin1 and promotes endocytosis